# ReviveEdit: Robust sequential editing via dominant subspace preservation

## Abstract

Sequential knowledge editing in large language models often causes catastrophic collapse of the model's general abilities, particularly for parameter-modifying methods. Existing approaches attempt to mitigate this issue with heuristic constraints, but they lack a principled understanding of the underlying failure mechanism and overlook the structured impact of edits on model parameters. In this work, we conduct a spectral analysis and identify a key failure mechanism: the progressive corruption of the dominant singular subspace of weight matrices, a low-rank subspace that we show is both crucial for encoding general abilities and highly sensitive to perturbations. Based on this insight, we propose REVIVE, a novel plug-and-play framework that prevents model collapse by explicitly preserving this dominant subspace. REVIVE projects any given update onto the singular vector basis of the original weight matrix and removes all components that would interfere with the protected subspace. This allows new knowledge to be integrated through less critical directions without damaging the model's core structure. Extensive experiments show that REVIVE substantially outperforms existing methods, maintaining high editing efficacy and preserving general capabilities even under extreme sequences of up to $20,000$ edits.

## 1 Introduction

Large language models (LLMs) often generate outdated or incorrect information due to flawed pre-training data or evolving real-world knowledge (Cao et al., 2021; Mitchell et al., 2022a; Sriramanan et al., 2024). Knowledge editing (Meng et al., 2022b) addresses this issue by updating specific facts in a lightweight and targeted manner. In practice, updates occur frequently, motivating the study of **sequential editing**, where a model undergoes multiple edits over time (Fang et al., 2024; Jiang et al., 2025b). This setting requires not only high edit success but also preservation of the model's general abilities (Gu et al., 2024), which poses particular challenges for parameter-modifying approaches, which are the focus of this paper. To alleviate issues of forgetting and collapse, recent methods such as RECT (Gu et al., 2024), NSE (Jiang et al., 2025b), PRUNE (Ma et al., 2024), and AlphaEdit (Fang et al., 2024) impose constraints at different levels. Such methods are generally based on the locate-then-edit paradigm, which first identifies the location of knowledge storage before updating it, thus making the research focus on how to modify the located matrix $\mathbf{W}$.

Despite demonstrating some effectiveness, the performance of existing methods remains unsatisfactory under cumulative updates. As shown in Figure 1, mainstream methods exhibit steadily declining effectiveness as the number of edits increases. We argue this is because existing methods largely overlook how the update matrix $\Delta\mathbf{W}$ precisely interacts with the original parameter matrix $\mathbf{W}$. This oversight impedes the ability to control adverse side effects from individual edits, ultimately leading to model collapse under cumulative updates. Specifically, methods such as RECT reduce harmful updates by thresholding parameter magnitudes. However, due to the black-box nature of LLMs, merely

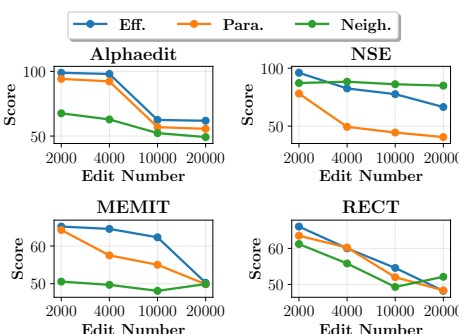

Figure 1: Results of current methods editing CounterFact with LLaMA3.

constraining the scale of $\Delta\mathbf{W}$ provides little insight into how general abilities or factual are actually affected. Neuron-level approaches like NSE modify specific neurons, but since individual neurons often encode entangled information, it remains challenging to precisely update facts without degrading general capabilities. Matrix-level constraints, as employed by PRUNE, regulate the condition number of $\Delta\mathbf{W}$ but fail to provide a fine-grained characterization of the modified matrix $\mathbf{W}+\Delta\mathbf{W}$. Similarly, AlphaEdit projects updates into the null space of key input vectors to localize changes; however, this constraint is defined in the input activation space, not the intrinsic parameter space. Consequently, its updates can still unintentionally disrupt the fundamental structure of the weight matrix. None of these methods, therefore, possess a systematic mechanism to analyze how edits interact with the intrinsic structure of the original parameters, limiting their effectiveness in preventing collapse during sequential editing.

To address the issue of model collapse under the parameter-modifying paradigm in sequential editing, we first conduct an in-depth analysis of the mainstream parameter-modifying methods (§ 2). Through spectral analysis of parameter matrices (§ 2.1) and preliminary experiments (§ 2.2), we find that the general abilities of LLMs are closely associated with subspaces spanned by the dominant singular values and their corresponding vectors. As edits accumulate, these dominant components are progressively perturbed, resulting in reduced editing success and impaired general ability. Based on these observations, we hypothesize that **model collapse in sequential editing is primarily caused by noise interfering with high-singular-value directions of weight matrices, which are essential for encoding general abilities.**

Building on this insight, we develop a **R**obust s**E**quential editing **V**ia dom**I**nant subspace preser**V**ation fram**E**work (REVIVE) (§ 3), which explicitly preserves the subspace spanned by the dominant singular values and vectors during sequential updates to prevent model collapse. Specifically, the key idea is to align all updates with the singular vector basis of the original weight matrix, enabling fine-grained decomposition of how edits interact with intrinsic functional directions. Based on this representation, REVIVE identifies the dominant subspace via a spectral energy criterion and constructs safe updates by filtering out components that interfere with this critical region. In this way, REVIVE preserves the high singular directions essential for general abilities while still allowing factual knowledge to be integrated over long editing sequences, thereby avoiding the cumulative degradation observed in existing methods.

Our contributions can be summarized as follows:

- We empirically establish that a key mechanism behind model collapse in sequential editing is the interference of updates with the dominant singular subspace of the original parameter matrices.

- We introduce a novel plug-and-play framework REVIVE that explicitly preserves the subspace spanned by the dominant singular values and singular vectors. This enables fine-grained modeling of how update matrices affect the original parameters, thereby ensuring that the model's general abilities are preserved during consecutive edits.

- We conduct extensive experiments on multiple models and benchmarks, demonstrating that REVIVE consistently and substantially outperforms state-of-the-art methods in both editing efficacy and the preservation of a model's general abilities.

## 2   WHY SEQUENTIAL EDITING COLLAPSES: A SPECTRAL PERSPECTIVE

To understand why sequential editing leads to model collapse, this section presents a spectral analysis of parameter matrices. As mainstream editing methods primarily target feed-forward network (FFN) layers for modification(Meng et al., 2022b;a), we ground our investigation in the FFN matrices of LLaMA3-8B. We first establish a view of each weight matrix as a composition of independent input-output mappings derived from its Singular Value Decomposition (SVD). This perspective allows us to investigate two critical questions: 1) Where are the model's general abilities concentrated within these mappings? and 2) How robust are these crucial components to perturbation? We then empirically demonstrate how existing editing methods, like MEMIT, progressively distort these mappings, leading to performance degradation. These analyses provide the foundation for our central hypothesis: **sequential editing fails because the cumulative noise from updates corrupts the dominant singular directions of weight matrices, which are essential for encoding general abilities.**

## 2.1 SPECTRAL VIEW OF PARAMETER MATRICES AS INPUT-OUTPUT MAPPINGS

From a spectral perspective, a parameter matrix $\mathbf{W} \in \mathbb{R}^{m \times n}$ can be decomposed into a set of independent input-output mappings using Singular Value Decomposition (SVD):

$$\mathbf{W} = \mathbf{U}\Sigma\mathbf{V}^\top = \sum_{i=1}^{r} \sigma_i \mathbf{u}_i \mathbf{v}_i^\top, \tag{1}$$

where $\mathbf{U} \in \mathbb{R}^{m \times m}$ and $\mathbf{V} \in \mathbb{R}^{n \times n}$ contain the orthogonal left and right singular vectors, respectively, and $\Sigma \in \mathbb{R}^{m \times n}$ contains the singular values $\sigma_1 \geq \sigma_2 \geq \cdots \geq \sigma_r \geq 0$. Each rank-one component $\sigma_i \mathbf{u}_i \mathbf{v}_i^\top$ acts as a distinct input-output mapping: an input vector $\mathbf{x} \in \mathbb{R}^n$ is projected onto $\mathbf{v}_i$, scaled by $\sigma_i$, and expanded along $\mathbf{u}_i$ to produce the output. The orthogonality of the singular vectors ensures these mappings operate independently. Pretraining learns this highly structured functional decomposition, making it a critical component of the model's general abilities(Wang et al., 2024b). Consequently, parameter-modifying methods (Meng et al., 2022a;b) that alter these matrices risk disrupting this fundamental structure.

## 2.2 CONCENTRATION AND ROBUSTNESS OF GENERAL ABILITIES

The spectral view raises two key questions: *where are general abilities concentrated among these mappings*, and *how robust are these components under perturbations?* We address these through two targeted experiments.

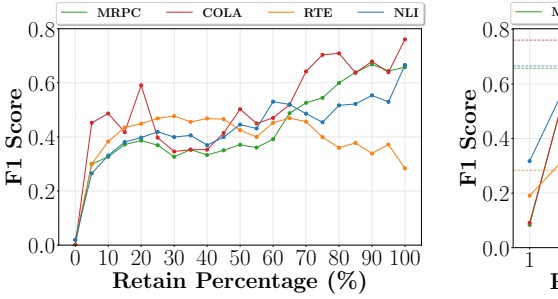

(a) General ability recovery with increasing proportion of retained singular components.

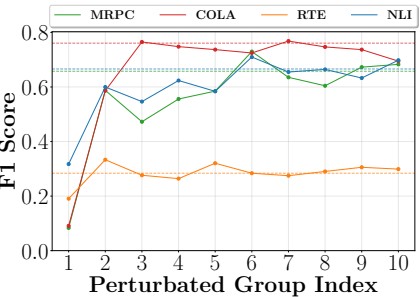

(b) Sensitivity of general ability to perturbations across different spectral groups.

Figure 2: Spectral concentration and fragility of general abilities.

### 2.2.1 CONCENTRATION OF GENERAL ABILITIES.

To locate where general abilities reside, we evaluate model performance on GLUE tasks (MRPC, CoLA, RTE, NLI) (Wang et al., 2019) using weight matrices reconstructed from a subset of their singular components. We define the singular value energy of an index set $\mathcal{I}$ as $E_{\mathcal{I}} = \sum_{i \in \mathcal{I}} \sigma_i$ and reconstruct $\mathbf{W}$ using the top components that capture $n\%$ of the total energy, $E_{\text{total}} = \sum_{i=1}^{r} \sigma_i$. This reconstruction is given by: $\tilde{\mathbf{W}}^n = \sum_{i \in \mathcal{I}} \sigma_i \mathbf{u}_i \mathbf{v}_i^\top$.

**Finding: General abilities are highly concentrated in the dominant singular subspace.** As shown in Figure 2a, reconstructing weight matrices with just the top $5\%$ of singular components (by energy) is sufficient to recover about $62.6\%$ of the model's original accuracy. This finding confirms that a model's general capabilities are encoded within a small, low-rank subspace which is spanned by the singular vectors corresponding to a few of the largest singular values.

### 2.2.2 ROBUSTNESS OF DOMINANT SUBSPACE MAPPINGS.

To evaluate the robustness of different singular components, we partition the singular components into ten non-overlapping groups by cumulative energy (0–10%, ..., 90–100%). For each group $\mathcal{G}$, we inject a structured rank-one perturbation. First, a random perturbation matrix is generated:

$$\mathbf{\Delta} = \sum_{j \in \mathcal{G}} \sum_{i=1}^{r} \alpha_{i,j} \, \mathbf{u}_i \mathbf{v}_j^\top, \qquad \alpha_{i,j} \sim \mathcal{N}(0, 1). \tag{2}$$

This matrix randomly remaps the input directions $\{v_j\}_{j\in G}$ to output directions. The perturbation is then normalized and scaled to a fixed strength: $\varepsilon$: $\tilde{\boldsymbol{\Delta}} = \varepsilon \cdot \frac{\boldsymbol{\Delta}}{\|\boldsymbol{\Delta}\|_F}$. This ensures all perturbations have an equal Frobenius norm, allowing for a fair comparison. We then measure the impact of perturbed matrix $\mathbf{W}' = \mathbf{W} + \tilde{\boldsymbol{\Delta}}$ on the model's general performance. A symmetric analysis on output directions is in the Appendix F.1 and shows similar trends.

**Finding: Modes associated with large singular values are highly sensitive to perturbations.**
As shown in Figure 2b, perturbations to the high-energy singular components (e.g., 0–20%) cause sharp and consistent degradation in performance. In contrast, perturbing the low-energy groups (70–100%) have only minor or negligible effects. These results reveal a paradoxical property: **the subspaces associated with the largest singular values, which are most crucial for general ability, are also the ones most susceptible to perturbations.** This fragility explains why indiscriminate parameter updates in sequential editing can easily disrupt general ability by corrupting the high-energy singular modes.

## 2.3 HOW SEQUENTIAL EDITING CORRUPTS THE DOMINANT SUBSPACE

Having established that general abilities are concentrated in a fragile and dominant singular subspace, we now analyze how these critical subspaces degrade during sequential editing progress. We introduce two spectral metrics and apply $2,000$ edits from COUNTERFACT dataset to LLaMA3 using MEMIT in 20 rounds (100 edits per round). We also conducted the same analytical experiments on AlphaEdit; results are deferred to Appendix F.2 due to space constraints. After each round, we evaluate the model's editing performance on key metrics (Efficacy Score and Paraphrase Score), its general abilities on the GLUE benchmark, and our spectral metrics, in order to investigate the **correlation** between overall model performance and internal parameters changes.

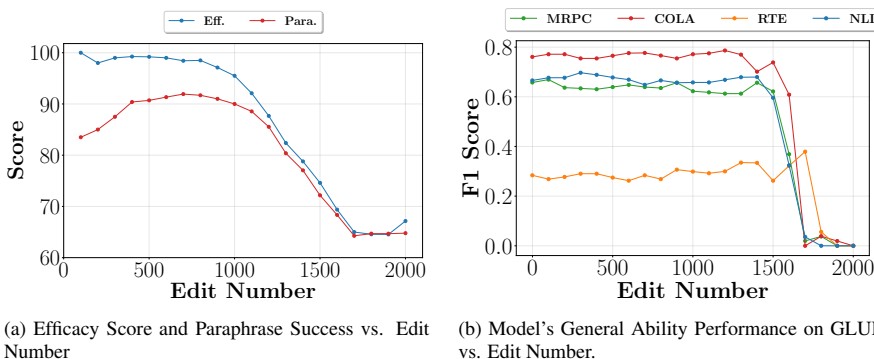

(a) Efficacy Score and Paraphrase Success vs. Edit Number

(b) Model's General Ability Performance on GLUE vs. Edit Number.

Figure 3: Performance collapse during sequential editing.

**Spectrum-based metrics.** We measure the stability of the dominant subspace (top 10% components by singular value energy) at both macroscopic and microscopic levels.

- **Low-rank Subspace Similarity (LS)** measures the macroscopic drift of the entire dominant singular subspace. It is the cosine similarity between the reconstructed low-rank approximations of the original matrix $\hat{\mathbf{W}}_0$ and the edited matrix $\hat{\mathbf{W}}_t$ (where $t$ denotes the editing round):

$$\text{LS}_t = \frac{\langle \hat{\mathbf{W}}_t, \hat{\mathbf{W}}_0 \rangle_F}{\|\hat{\mathbf{W}}_t\|_F \cdot \|\hat{\mathbf{W}}_0\|_F}. \tag{3}$$

- **Singular Vector Similarity (SS)** provides a microscopic view by measuring how individual dominant singular vectors rotate. We compute the cosine similarity between a dominant singular vector $\mathbf{v}_t$ from the edited matrix $\mathbf{W}_t$ and every original singular vector $\mathbf{v}_j$ from $\mathbf{W}_0$(a vector basis): $\text{SS}_t^j = \langle \mathbf{v}_t, \mathbf{v}_j \rangle$. Results for left singular vectors show the same trend, see Appendix F.3.

**Finding: Sequential editing causes model collapse precisely because it progressively corrupts the dominant singular subspace.** The evidence for this connection is clear across all levels of analysis. At the *behavioral level* (Figure 3a, 3b), both edit success and general ability decline

steadily after round 10, collapsing almost completely by round 20. This performance degradation is perfectly tracked at the *macroscopic level* by our Low-rank Subspace Similarity (LS) metric (Figure 4), which remains high initially before drifting and declining sharply after round 15. The *microscopic* cause of this drift is revealed by our Singular Vector Similarity (SS) metric (Figure 5), which shows that individual dominant singular vectors steadily rotate away from their original directions, becoming nearly orthogonal by round 20. This signifies a fundamental corruption of the learned input-output mappings. Together, these results provide strong evidence that model collapse is structurally rooted in the degradation of the dominant singular subspace.

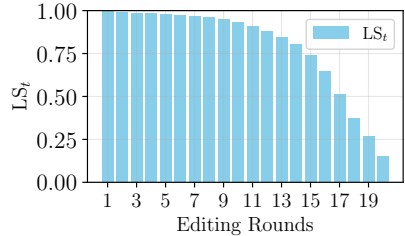

Figure 4: $LS_t$ vs. Editing Rounds

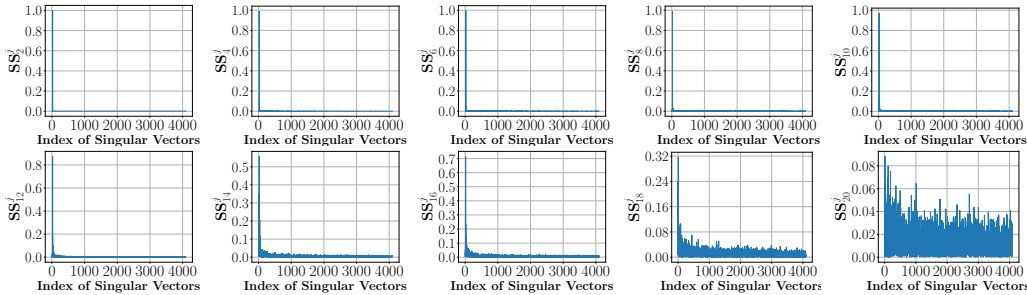

Figure 5: Singular Vector Similarity (SS) across sequential edits (rounds 2-20, every 2 rounds).

## 3 METHODOLOGY

Our spectral analysis in Section 2 reveals a key mechanism behind model collapse in sequential editing: the cumulative corruption of the dominant singular subspace, which encodes the model's general abilities. Motivated by this finding, we propose **R**obust s**E**quential editing **V**ia dom**I**nant subspace preser**V**ation fram**E**work (REVIVE), designed to directly counteract this failure mechanism. The overall architecture is illustrated in Figure 6. The core idea of REVIVE is to represent and constrain edits within the singular vector basis of the original weight matrix. This allows us to first identify the dominant subspace critical for general abilities and then construct a safe update by surgically removing any components that would interfere with this protected region. The full algorithm is detailed in Appendix D.

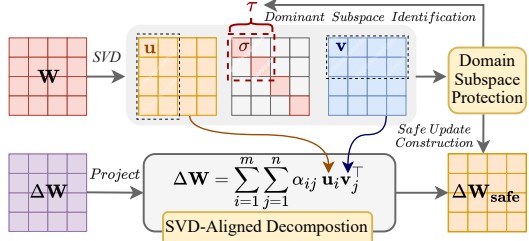

Figure 6: An overview of the REVIVE. An arbitrary update matrix $\Delta\mathbf{W}$ is first projected onto the SVD basis of the original weight matrix $\mathbf{W}$. The dominant subspace, identified via an energy threshold $\tau$, is then used to filter the projected update, resulting in a safe update $\Delta\mathbf{W}_{\text{safe}}$ that avoids corrupting the model's core structure.

### 3.1 SVD-ALIGNED DECOMPOSITION OF UPDATES

The foundation of our approach is to analyze any update matrix $\Delta\mathbf{W}$ within the intrinsic coordinate system defined by the original weight matrix $\mathbf{W}$. As shown in Equation (1), the SVD of $\mathbf{W}$ provides its left and right singular vectors, $\{\mathbf{u}_i\}$ and $\{\mathbf{v}_j\}$. The set of their rank-one outer products, $\{\mathbf{u}_i\mathbf{v}_j^\top\}_{ij}$, forms a complete orthogonal basis for the matrix space $\mathbb{R}^{m\times n}$ (see Appendix C). We project an arbitrary update matrix $\Delta\mathbf{W}$, generated by any editing method, onto this SVD basis to decompose its effect along each of the original matrix's functional directions:

$$\Delta\mathbf{W} = \sum_{i=1}^{m}\sum_{j=1}^{n} \alpha_{ij}\, \mathbf{u}_i\mathbf{v}_j^\top. \tag{4}$$

This decomposition provides a fine-grained view of the update. The coefficients $\alpha_{ij} = \langle \Delta \mathbf{W}, \mathbf{u}_i \mathbf{v}_j^\top \rangle_F$ precisely quantify how much the edit affects the mapping from each original input direction $\mathbf{v}_j$ to each original output direction $\mathbf{u}_i$. This representation is key to precisely controlling the update's impact.

## 3.2 Dominant Subspace Protection

The structure of $\mathbf{W}$ is dominated by its larger singular values, which capture the most critical and stable general ability learned during pretraining. Sequential editing, however, often introduces nonzero $\alpha_{ij}$ aligned with these dominant subspace mappings. While a single perturbation may have little effect, their accumulation over long editing sequences erodes the high-singular subspace and leads to collapse. Our proposed Dominant Subspace Protection (DSP) module counters this problem through two steps: identifying the dominant subspace and constructing safe updates.

**Dominant Subspace Identification.** To identify the critical components for preservation, we adopt an energy-based criterion. Specifically, we define a **singular-value energy threshold** $\tau \in (0, 1)$ (the impact of $\tau$ can be found in Section 4.2.) and select the smallest index $k$ such that the cumulative energy of the top-$k$ singular values exceeds this threshold:

$$\frac{\sum_{i=1}^{k} \sigma_i}{\sum_{i=1}^{r} \sigma_i} \geq \tau \tag{5}$$

The singular vectors corresponding to these top-k singular values, $\{\mathbf{u}_i\}_{i=1}^{k}$ and $\{\mathbf{v}_i\}_{i=1}^{k}$, span the dominant input and output subspaces.

**Safe Update Construction.** Once the dominant subspace is identified, we construct a *safe update* by removing all components of $\Delta \mathbf{W}$ associated with the dominant singular vectors. Concretely, we set any coefficient $\alpha_{ij}$ to zero if its its corresponding input vector $\mathbf{v}_j$ or output vector $\mathbf{u_i}$ is part of the dominant subspace (i.e., if $j \leq k$ or $i \leq k$). The resulting safe update matrix, $\Delta \mathbf{W}_{\text{safe}}$, contains only components that operate outside of this protected region:

$$\Delta \mathbf{W}_{\text{safe}} = \sum_{i>k} \sum_{j>k} \alpha_{ij} \, \mathbf{u}_i \mathbf{v}_j^\top. \tag{6}$$

This operation ensures that any modification is realized exclusively through low-energy directions, thereby avoiding interference with the dominant components that are essential for preserving general abilities.

By explicitly shielding the dominant subspace from perturbation, REVIVE allows factual knowledge to be continuously integrated without corrupting the core components responsible for the model's general abilities. As a result, the model maintains stability across long editing sequences and avoids the cumulative degradation that typically leads to catastrophic collapse.

## 4 Experiments

### 4.1 Experiment Setup

**Base Models.** We conduct experiments on three widely adopted LLMs in the knowledge editing: GPT2-XL (1.5B) (Radford et al., 2019), GPT-J (6B) (Wang & Komatsuzaki, 2021), and LLaMA3 (8B) (Grattafiori et al., 2024).

**Baselines.** We compare REVIVE against a suite of strong baselines, including the canonical MEMIT (Meng et al., 2022b) , as well as four state-of-the-art methods designed for sequential editing: ALPHAEDIT (Fang et al., 2024), PRUNE (Ma et al., 2024), RECT (Gu et al., 2024), and NSE (Jiang et al., 2025b). Further details and comparisons are available in Appendix E.1.

**Datasets and Metrics.** We use two standard factual knowledge editing benchmarks, COUNTER-FACT (Meng et al., 2022b) and ZsRE (Levy et al., 2017). For ZsRE, we measure Efficacy, Paraphrase, and Neighborhood Scores. For COUNTERFACT, we add Fluency and Consistency metrics. Detailed definitions are provided in Appendix E.2, E.3, and E.4.

Table 1: Performance on sequential editing over 10,000 Samples. The abbreviations *Eff.* (Efficacy Success), *Para.* (Paraphrase Success), *Neigh.* (Neighborhood Success), *Flu.* (Generation Entropy), and *Consis.* (Reference Score) denote respective evaluation metrics. Relative improvements (%) are shown in blue and decreases in orange. ↑↑ indicates a large improvement where the baseline score was near zero.

| Method | Counterfact | | | | | ZsRE | | |
|---|---|---|---|---|---|---|---|---|
| | Eff.↑ | Para.↑ | Neigh.↑ | Flu.↑ | Consis.↑ | Eff.↑ | Para.↑ | Neigh.↑ |
| **LLaMA3** | 7.02 | 9.44 | 89.73 | 635.47 | 24.24 | 35.67 | 34.81 | 31.83 |
| MEMIT | 62.3 | 55.02 | 48.11 | 522.1 | 4.4 | 0.08 | 0.08 | 1.36 |
| +REVIVE | 95.62 ↑53.5% | 84.60 ↑53.8% | 62.17 ↑29.2% | 603.22 ↑15.5% | 29.39 ↑568.0% | 83.45 ↑↑ | 79.90 ↑↑ | 32.01 ↑↑ |
| PRUNE | 59.98 | 55.72 | 48.56 | 571.27 | 1.89 | 0.00 | 0.00 | 0.08 |
| +REVIVE | 80.57 ↑34.3% | 69.54 ↑24.7% | 54.76 ↑12.8% | 570.85 ↓0.1% | 28.49 ↑↑ | 56.61 ↑↑ | 53.30 ↑↑ | 27.74 ↑↑ |
| RECT | 60.23 | 54.9 | 50.56 | 441.61 | 5.08 | 0.00 | 0.00 | 0.00 |
| +REVIVE | 92.69 ↑53.9% | 79.95 ↑45.6% | 63.09 ↑24.8% | 600.13 ↑35.9% | 29.28 ↑476.8% | 84.20 ↑↑ | 80.27 ↑↑ | 31.92 ↑↑ |
| AlphaEdit | 62.48 | 56.9 | 52.31 | 505.5 | 4.25 | 90.57 | 85.66 | 30.5 |
| +REVIVE | 98.74 ↑58.0% | 90.08 ↑58.4% | 60.19 ↑15.1% | 615.97 ↑21.9% | 32.66 ↑668.5% | 93.40 ↑3.1% | 89.31 ↑4.3% | 31.72 ↑4.0% |
| NSE | 77.59 | 44.42 | 86.12 | 607.86 | 23.31 | 45.61 | 45.04 | 31.27 |
| +REVIVE | 98.89 ↑27.4% | 92.28 ↑107.8% | 65.72 ↓23.6% | 618.66 ↑1.8% | 32.74 ↑40.5% | 94.37 ↑107.0% | 90.57 ↑101.2% | 32.17 ↑2.9% |
| **GPT-J** | 15.22 | 17.65 | 83.50 | 622.01 | 29.61 | 26.45 | 25.74 | 27.04 |
| MEMIT | 54.03 | 52.66 | 53.63 | 594.16 | 5.17 | 0.10 | 0.10 | 0.17 |
| +REVIVE | 97.63 ↑80.7% | 87.76 ↑66.6% | 66.52 ↑24.1% | 616.47 ↑3.8% | 40.69 ↑687.4% | 88.88 ↑↑ | 83.22 ↑↑ | 27.87 ↑↑ |
| PRUNE | 52.92 | 51.47 | 53.91 | 576.95 | 5.14 | 0.03 | 0.02 | 0.05 |
| +REVIVE | 86.95 ↑64.3% | 81.03 ↑57.5% | 64.21 ↑19.1% | 583.05 ↑1.1% | 35.73 ↑595.7% | 63.08 ↑↑ | 58.90 ↑↑ | 26.03 ↑↑ |
| RECT | 63.60 | 55.33 | 56.69 | 404.13 | 4.49 | 23.60 | 22.02 | 12.44 |
| +REVIVE | 94.96 ↑49.4% | 77.27 ↑39.6% | 67.78 ↑19.6% | 612.76 ↑244.4% | 38.69 ↑761.7% | 81.28 ↑244.4% | 74.78 ↑239.6% | 28.20 ↑126.7% |
| AlphaEdit | 96.51 | 86.76 | 60.80 | 544.18 | 19.33 | 87.84 | 78.65 | 22.31 |
| +REVIVE | 99.50 ↑3.1% | 93.92 ↑8.3% | 67.35 ↑10.8% | 600.64 ↑10.4% | 40.63 ↑110.3% | 97.53 ↑11.0% | 91.33 ↑16.1% | 23.40 ↑4.9% |
| NSE | 88.95 | 69.69 | 75.46 | 611.35 | 33.31 | 44.03 | 42.39 | 24.86 |
| +REVIVE | 94.88 ↑6.7% | 89.49 ↑28.4% | 64.06 ↓15.1% | 608.12 ↓0.5% | 40.18 ↑20.6% | 97.80 ↑122.1% | 91.75 ↑116.4% | 26.84 ↑8.0% |
| **GPT2-XL** | 21.82 | 24.16 | 78.32 | 626.69 | 31.34 | 22.17 | 21.28 | 24.20 |
| MEMIT | 70.56 | 62.42 | 55.94 | 516.26 | 8.74 | 53.00 | 46.27 | 12.76 |
| +REVIVE | 90.46 ↑28.2% | 75.88 ↑21.5% | 63.83 ↑14.1% | 598.21 ↑15.9% | 34.32 ↑292.7% | 68.20 ↑28.7% | 60.80 ↑31.4% | 27.09 ↑112.4% |
| PRUNE | 57.61 | 54.01 | 52.87 | 596.56 | 6.93 | 0.21 | 0.19 | 2.06 |
| +REVIVE | 82.00 ↑42.4% | 70.90 ↑31.3% | 62.82 ↑18.8% | 600.99 ↑0.7% | 34.55 ↑398.1% | 40.92 ↑↑ | 37.61 ↑↑ | 25.29 ↑1127.2% |
| RECT | 86.52 | 69.50 | 55.71 | 499.64 | 11.41 | 29.80 | 27.17 | 6.94 |
| +REVIVE | 82.99 ↓4.1% | 69.20 ↓0.4% | 65.60 ↑17.7% | 595.69 ↑19.2% | 34.05 ↑198.3% | 62.45 ↑109.6% | 55.17 ↑103.0% | 26.20 ↑278.0% |
| AlphaEdit | 92.13 | 76.80 | 56.85 | 581.49 | 31.72 | 53.00 | 46.27 | 12.76 |
| +REVIVE | 94.48 ↑2.6% | 78.70 ↑2.5% | 62.87 ↑10.6% | 587.94 ↑1.1% | 38.51 ↑21.5% | 68.10 ↑28.5% | 57.17 ↑23.5% | 20.35 ↑59.4% |
| NSE | 69.22 | 54.54 | 69.26 | 596.41 | 28.87 | 33.71 | 32.31 | 22.70 |
| +REVIVE | 96.12 ↑38.8% | 84.49 ↑54.9% | 64.17 ↓7.4% | 592.71 ↓0.6% | 37.74 ↑30.9% | 77.83 ↑131.0% | 70.55 ↑118.4% | 24.84 ↑9.4% |

## 4.2 Results and Analysis

This section presents a comprehensive evaluation of REVIVE. We first demonstrate its effectiveness in a 10,000-edit sequential task, assessing both editing success and the preservation of general abilities. We then conduct further analyses of its robustness, including hyperparameter sensitivity, scalability to 20,000 edits, and a visualization of its ability to preserve representational structure.

**Sequential Editing Performance.** To validate the effectiveness of our REVIVE in sequential editing, we evaluate REVIVE's performance over an extended sequence of 10,000 edits, applied in 100 rounds of 100 edits each, on the Counterfact and ZsRE benchmarks. As shown in Table 1, applying REVIVE as a plug-and-play module leads to substantial and consistent performance gains across all base methods and models. The most dramatic improvements occur on the challenging ZsRE dataset, where methods like MEMIT and RECT quickly collapse to near-zero performance

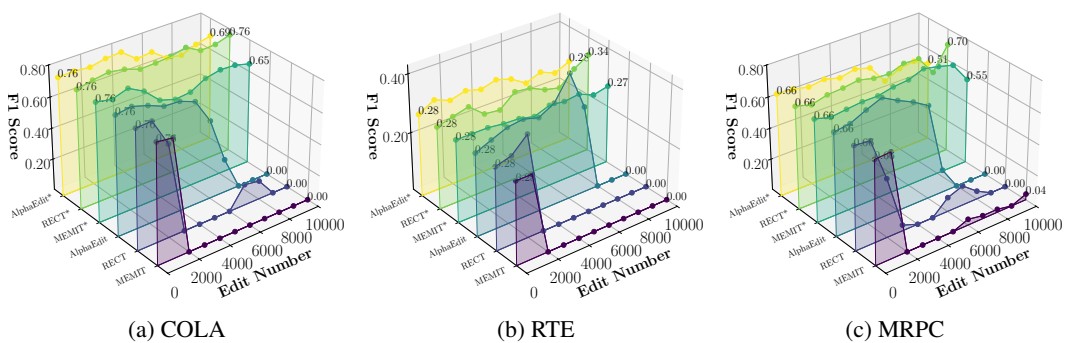

Figure 7: Performance of baselines and their REVIVE-enhanced versions (*) on GLUE datasets.

on their own. With REVIVE, however, they are not only stabilized but achieve high efficacy scores (e.g., 83.45% for MEMIT+REVIVE on LLaMA3), demonstrating that our method can rescue baselines from complete failure. Notably, the standard MEMIT+REVIVE combination consistently surpasses specialized sequential baselines like PRUNE and RECT, suggesting that proactively protecting the dominant subspace is a more effective strategy than post-hoc constraints. We note that for some methods like NSE, applying REVIVE leads to a numerical decrease in Neighborhood Success. This is likely because the baseline's high score is an artifact of its low editing efficacy; an edit that fails to modify the model will trivially preserve neighborhood consistency. Therefore, REVIVE's ability to achieve massive gains in Efficacy and Paraphrase while keeping Neighborhood Success high represents a more genuine and robust form of successful editing.

**Preservation of General Abilities.**    To evaluate the ability of our method to preserve general abilities, we assess how well REVIVE preserves general abilities by evaluating the edited LLaMA3 model on the GLUE benchmark after 10,000 sequential edits. For brevity, we present results on three representative datasets in Figure 7 and include full results in Appendix F.4, as all datasets show consistent trends. As shown in Figure 7,baseline methods without protection degrade rapidly. MEMIT and RECT collapse to near-zero performance after only 3,000 edits, and even the more robust ALPHAEDIT eventually suffers a complete collapse after 8,000 edits. In contrast, REVIVE enhanced methods maintains an overall average 86.34% of its performance across all tasks after 10,000 edits. These results clearly demonstrate that shielding the dominant singular subspace is a highly effective strategy for preserving a model's general abilities during sequential editing.

**Sensitivity Analysis.**    We evaluate the stability of REVIVE by analyzing its sensitivity to its single intrinsic hyperparameter, the singular value energy threshold. This parameter ($\tau$) is defined in Section 3.2, controls the size of the dominant subspace shielded from edits. A higher $\tau$ better preserves the model's fragile general abilities but may limit edit capacity, while a lower $\tau$ allows for more impactful edits at the risk of corrupting critical singular directions. Figure 8 shows that REVIVE exhibits strong robustness, maintaining high performance across a wide range of $\tau$ values. This stability indicates that our method is not sensitive to the exact delineation of the dominant subspace and removes the need for costly hyperparameter tuning. Further hyperparameter results for all models are in Appendix F.5, and

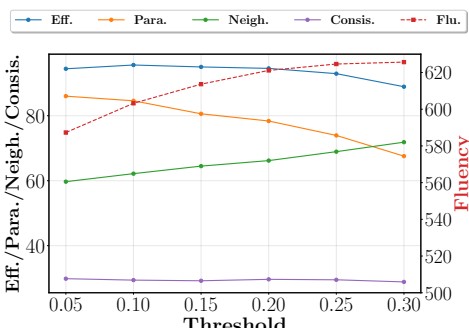

Figure 8: Performance of MEMIT-REVIVE on LLaMA3 (CounterFact) under different thresholds.

an additional analysis of batch size impact is provided in Appendix F.6.

**Scalability under Extreme Sequential Editing.**    To stress-test the scalability of REVIVE, we conduct experiments at a significantly larger scale on LLaMA3. We apply 20,000 sequential edits on COUNTERFACT (in 200 rounds of 100) and the full 19,086 edits on ZsRE. As shown in Figure 9, REVIVE continues to deliver substantial gains over the original base methods, averaging +75.1%

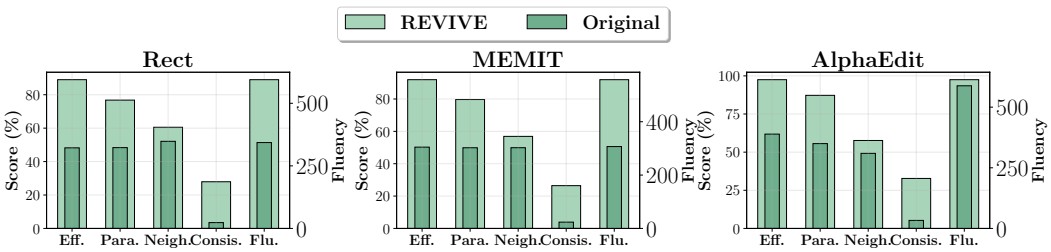

Figure 9: REVIVE vs. original methods under 20,000 sequential edits on COUNTERFACT.

in Efficacy and +53.1% in Fluency on Counterfact. Complete results for all metrics and methods, provided in Appendix F.7, demonstrate that our module effectively maintains editing performance even when the number of edits is significantly scaled up.

**Visualization of Representational Stability.** To visually inspect how sequential editing affects the model's internal geometry, we use t-SNE (Maaten & Hinton, 2008) to visualize the representations of 1,000 factual prompts from LLaMA3, both before and after applying 20,000 edits. As illustrated in Figure 10, a strong baseline like ALPHAEDIT causes a

noticeable distributional shift, where post-edit representations drift away from their original positions. In contrast, MEMIT+REVIVE keeps the post-edit representations tightly clustered with their original counterparts. This visualization offers powerful qualitative evidence for our core claim: by preserving the dominant subspace, REVIVE maintains not just downstream performance but also the fundamental representational structure of the model.

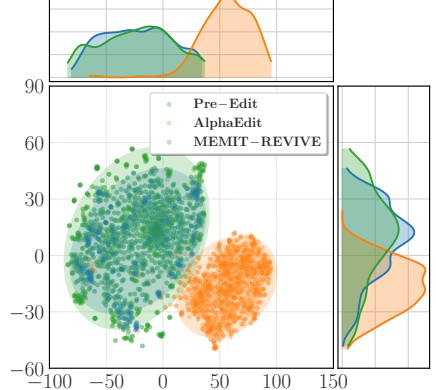

Figure 10: t-SNE visualization of representations after 20,000 sequential edits on LLaMA3.

## 5 RELATED WORK

Our work focuses on parameter-modifying methods for knowledge editing. While early approaches (Mitchell et al., 2022a; Meng et al., 2022b) are effective for single edits, they often fail in sequential scenarios due to accumulating interference. To mitigate this, recent methods have introduced various heuristic constraints, such as enforcing sparsity (RECT (Gu et al., 2024)), controlling update condition numbers (PRUNE (Ma et al., 2024)), or projecting into a null space (ALPHAEDIT (Fang et al., 2024)). These approaches, however, target symptoms of degradation rather than the underlying cause. In contrast, our method is based on a spectral analysis that identifies the root cause of collapse as the corruption of dominant functional subspaces, and it intervenes directly to preserve them. A comprehensive review of the field is provided in Appendix E.7.

## 6 CONCLUSION

In this work, we investigated the critical challenge of model collapse in sequential knowledge editing. We conducted a spectral analysis that identified a key failure mechanism: the cumulative corruption of the dominant singular subspace of weight matrices, which is essential for preserving a model's general abilities. To counteract this, we introduced REVIVE, a plug-and-play framework that safeguards this critical subspace. By projecting updates onto the SVD basis of the original weights and removing components that interfere with the dominant subspace, REVIVE allows for robust and scalable editing. Extensive experiments confirmed that our approach substantially improves editing efficacy and preserves general abilities far better than existing methods, even under extreme editing scenarios. Our findings provide a deeper, structural understanding of model collapse and offer a principled, effective solution to ensure the long-term stability of edited LLMs.

## ETHICAL CONSIDERATIONS

Our research focuses on improving the reliability of large language models by correcting factual inaccuracies, which is a beneficial application of knowledge editing. The methods developed are intended to enhance the safety and trustworthiness of AI systems. However, we acknowledge that any model editing technology could potentially be misused for malicious purposes, such as injecting biased or harmful information. Our proposed method, REVIVE, is designed to be a general-purpose tool for stabilizing sequential edits and does not inherently favor any particular type of content. The responsibility for the nature of the edited content lies with the user applying the method. Furthermore, all datasets used in our experiments (COUNTERFACT, ZsRE, GLUE) are standard, publicly available academic benchmarks that have been widely vetted by the research community. We do not foresee any direct negative societal impacts stemming from our work.

## REPRODUCIBILITY STATEMENT

We are committed to ensuring the full reproducibility of our research. To this end, our source code, is provided in the supplementary materials. Our work relies exclusively on publicly available models (GPT2-XL, GPT-J, LLaMA3) and standard benchmarks (COUNTERFACT, ZsRE, GLUE), as detailed in our Experimental Setup (§E.1). The theoretical underpinnings of our method are described Section 3.1, with proofs provided in the Appendix C. All hyperparameters required to reproduce our main results are also detailed in the Appendix E.6, providing a clear path for the replication of our results.

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

## A  USAGE OF LARGE LANGUAGE MODELS

In the preparation of this work, large language models (LLMs) have been utilized to assist in several stages of writing. In particular, LLMs played a significant role in polishing the manuscript by improving readability and correcting grammatical issues. Moreover, they provided valuable assistance in certain aspects of data visualization, such as generating and refining plotting scripts, which streamlined the experimental analysis process.

## B    LIMITATIONS & FUTURE DISCUSSION

While our work demonstrates the effectiveness of preserving the dominant singular subspace, we acknowledge several limitations that open avenues for future research.

First, our method relies on a one-time SVD of the original weight matrices, which may bring additional computational and storage cost. Future work could explore more efficient, decomposition-free methods for identifying and protecting these critical subspaces.

Second, while REVIVE protects the subspace critical for general abilities, it does not guarantee every specific pieces of knowledge is preserved. This can lead to perturbation in metrics like Neighborhood Success, as observed with NSE. Designing a more precise, knowledge-aware subspace protection mechanism that distinguishes between general abilities and specific facts is a promising direction for future work.

Third, our analysis and experiments have primarily focused on the feed-forward network layers. While these are critical for knowledge storage, extending this spectral analysis to other components like attention mechanisms is an important next step.

## C. COMPARISON WITH EXISTING PROJECTION-BASED EDITING APPROACHES

In this section, we offer a detailed exposition of how the REVIVE framework differs from existing projection-based editing approaches. We further analyze the usability, computational effectiveness, and scalability of REVIVE within the singular-vector space, and articulate the considerations that motivate its post-hoc projection design.

**Difference from AlphaEdit.**    Although both methods involve applying projection to parameter updates, the underlying motivations are fundamentally different. AlphaEdit(Fang et al., 2024) assumes that sequential degradation is caused by knowledge interference and constructs a knowledge covariance matrix from 100k factual triples to extract a null space that avoids such interference. Its preserved subspace therefore comes from *external knowledge statistics*.

In contrast, our analysis identifies a different failure mode—*erosion of the dominant singular subspace* of FFN parameters—which encodes general model abilities. REVIVE thus protects the *intrinsic dominant subspace* derived from the model's own spectral structure. While both approaches use projections in form, the protected subspaces and the mechanisms they address are fundamentally different. This distinction also explains behavioral differences: AlphaEdit performs well early on but begins to deteriorate around 8k edits, whereas methods augmented with REVIVE remain stable beyond 20k edits.

**Difference from PRUNE.**    PRUNE(Ma et al., 2024) directly suppresses update singular values larger than the maximum singular value of the original parameter matrix, without distinguishing the directions associated with those singular values. This magnitude-only suppression cannot effectively preserve the model's functional subspace and may attenuate useful update components while allowing harmful ones to remain. In contrast, REVIVE explicitly preserves dominant singular directions and filters only components that would distort them, addressing a type of degradation that PRUNE is not designed to handle.

**Difference from Delta-Edit and O-Edit.**    Delta-Edit(Cao et al., 2025) and O-Edit(Cai & Cao, 2024) track directions of previous edits and project new updates to avoid overwriting past changes. Their protected subspaces are derived from *accumulated edit history* and address inter-edit interference. REVIVE targets a different failure mode: progressive corruption of the dominant singular subspace of FFN parameters (energy decay and directional rotation), which arises even when edits are unrelated. Accordingly, REVIVE preserves the dominant singular directions of the parameter matrix rather than historical edit directions.

## C   Proof of the SVD-aligned Matrix Basis

This section provides the formal proof for the claim made in Section 3.1 that the set of rank-one outer products $\{\mathbf{u}_i \mathbf{v}_j{}^\top\}_{i,j}$ derived from the singular vectors of a matrix $\mathbf{W}$, forms an orthonormal basis for the space of matrices $\mathbb{R}^{m \times n}$.

**Theorem 1** (Outer-product bases from two orthonormal vector bases). *Let $\{\mathbf{u}_1, \ldots, \mathbf{u}_m\} \subset \mathbb{R}^m$ and $\{\mathbf{v}_1, \ldots, \mathbf{v}_n\} \subset \mathbb{R}^n$ be orthonormal bases of $\mathbb{R}^m$ and $\mathbb{R}^n$ respectively. Consider the set of $mn$ matrices*

$$\mathcal{B} = \{ \mathbf{u}_p \mathbf{v}_q^\top \;:\; p = 1, \ldots, m, \; q = 1, \ldots, n \}.$$

*Then $\mathcal{B}$ forms an orthonormal basis of the real vector space $\mathbb{R}^{m \times n}$ with respect to the Frobenius inner product $\langle \mathbf{X}, \mathbf{Y} \rangle_F = \mathrm{tr}(\mathbf{X}^\top \mathbf{Y})$. In particular, every $\mathbf{X} \in \mathbb{R}^{m \times n}$ admits the unique expansion*

$$\mathbf{X} = \sum_{p=1}^{m} \sum_{q=1}^{n} c_{pq}\, \mathbf{u}_p \mathbf{v}_q^\top, \qquad c_{pq} = \langle \mathbf{X}, \mathbf{u}_p \mathbf{v}_q^\top \rangle_F = \mathbf{u}_p^\top \mathbf{X}\, \mathbf{v}_q.$$

*Proof.* We split the proof into three parts: (i) orthogonality, (ii) spanning (completeness), and (iii) uniqueness / coefficient formula.

**(i) Orthogonality.**   Take two generic elements $\mathbf{u}_p \mathbf{v}_q^\top$ and $\mathbf{u}_{p'} \mathbf{v}_{q'}^\top$ from $\mathcal{B}$. Their Frobenius inner product is

$$\langle \mathbf{u}_p \mathbf{v}_q^\top, \; \mathbf{u}_{p'} \mathbf{v}_{q'}^\top \rangle_F = \mathrm{tr}\big((\mathbf{u}_p \mathbf{v}_q^\top)^\top (\mathbf{u}_{p'} \mathbf{v}_{q'}^\top)\big) = \mathrm{tr}\big(\mathbf{v}_q \mathbf{u}_p^\top \mathbf{u}_{p'} \mathbf{v}_{q'}^\top\big).$$

By cyclicity of the trace,

$$\langle \mathbf{u}_p \mathbf{v}_q^\top, \; \mathbf{u}_{p'} \mathbf{v}_{q'}^\top \rangle_F = (\mathbf{u}_p^\top \mathbf{u}_{p'})(\mathbf{v}_q^\top \mathbf{v}_{q'}).$$

Since $\{\mathbf{u}_p\}$ and $\{\mathbf{v}_q\}$ are orthonormal bases, we have

$$\mathbf{u}_p^\top \mathbf{u}_{p'} = \delta_{pp'}, \qquad \mathbf{v}_q^\top \mathbf{v}_{q'} = \delta_{qq'},$$

where $\delta_{ij}$ is the Kronecker delta:

$$\delta_{ij} = \begin{cases} 1, & i = j, \\ 0, & i \neq j. \end{cases}$$

Therefore,

$$\langle \mathbf{u}_p \mathbf{v}_q^\top, \; \mathbf{u}_{p'} \mathbf{v}_{q'}^\top \rangle_F = \delta_{pp'} \delta_{qq'}.$$

In particular, if either $p \neq p'$ or $q \neq q'$, then one of the Kronecker deltas vanishes, making the inner product equal to $0$. This proves that distinct basis elements are orthogonal.

**(ii) Spanning (completeness).**   The vector space $\mathbb{R}^{m \times n}$ has dimension $mn$. We have produced $mn$ elements in $\mathcal{B}$ which are mutually orthonormal; mutual orthonormality implies linear independence. Because we have exactly $mn$ linearly independent matrices in an $mn$-dimensional space, $\mathcal{B}$ must span $\mathbb{R}^{m \times n}$, and therefore forms a basis.

For a constructive argument, let $\mathbf{X} \in \mathbb{R}^{m \times n}$ be arbitrary. Define coefficients

$$c_{pq} = \langle \mathbf{X}, \mathbf{u}_p \mathbf{v}_q^\top \rangle_F = \mathbf{u}_p^\top \mathbf{X}\, \mathbf{v}_q.$$

Form the matrix

$$\widehat{\mathbf{X}} = \sum_{p=1}^{m} \sum_{q=1}^{n} c_{pq}\, \mathbf{u}_p \mathbf{v}_q^\top.$$

For any fixed indices $(p', q')$ we compute

$$\langle \widehat{\mathbf{X}}, \mathbf{u}_{p'} \mathbf{v}_{q'}^\top \rangle_F = \sum_{p,q} c_{pq} \langle \mathbf{u}_p \mathbf{v}_q^\top, \mathbf{u}_{p'} \mathbf{v}_{q'}^\top \rangle_F = \sum_{p,q} c_{pq}\, \delta_{pp'} \delta_{qq'} = c_{p'q'}.$$

But by definition $c_{p'q'} = \langle \mathbf{X}, \mathbf{u}_{p'} \mathbf{v}_{q'}^\top \rangle_F$, hence

$$\langle \widehat{\mathbf{X}} - \mathbf{X}, \mathbf{u}_{p'} \mathbf{v}_{q'}^\top \rangle_F = 0 \quad \text{for all } p', q'.$$

Since $\mathcal{B}$ spans the space, the only matrix orthogonal to every basis element is the zero matrix; therefore $\widehat{\mathbf{X}} - \mathbf{X} = \mathbf{0}$, proving $\mathbf{X} = \widehat{\mathbf{X}}$. This provides an explicit expansion of any matrix in the basis $\mathcal{B}$, proving completeness.

**(iii) Uniqueness and coefficient formula.** Orthogonality gives immediately that the coefficients are unique and equal to the Frobenius inner products:

$$c_{pq} = \langle \mathbf{X}, \mathbf{u}_p \mathbf{v}_q^\top \rangle_F = \mathbf{u}_p^\top \mathbf{X} \mathbf{v}_q.$$

This completes the proof. Therefore, using the $\mathbf{u}$ and $\mathbf{v}$ matrices obtained from the SVD of a matrix to construct such outer-product basis matrices is valid and well-founded. $\square$

# D  ALGORITHM DETAILS

---

**Algorithm 1** REVIVE

---

**Require:** Current weight matrix $\mathbf{W} \in \mathbb{R}^{m \times n}$; update matrix $\Delta \mathbf{W}$; singular-value energy threshold $\tau \in (0, 1)$

**Ensure:** Safe update $\Delta \mathbf{W}_{\text{safe}}$

1: **SVD-Aligned Decomposition:**
2: $\{\mathbf{u}_i\}_{i=1}^m, \{\sigma_i\}_{i=1}^r, \{\mathbf{v}_i\}_{i=1}^n = \text{SVD}(\mathbf{W})$
3: Construct orthogonal basis $\{\mathbf{u}_i \mathbf{v}_j^\top \mid i = 1, \ldots, m; j = 1, \ldots, n\}$
4: **for** $i = 1$ to $m$ **do**
5:     **for** $j = 1$ to $n$ **do**
6:         $\alpha_{ij} \leftarrow \langle \Delta \mathbf{W}, \mathbf{u}_i \mathbf{v}_j^\top \rangle_F$
7:     **end for**
8: **end for**
9: Represent update as $\Delta \mathbf{W} = \sum_{i=1}^m \sum_{j=1}^n \alpha_{ij} \mathbf{u}_i \mathbf{v}_j^\top$
10: **Dominant Subspace Identification:**
11: Find smallest $k$ s.t. $\frac{\sum_{i=1}^k \sigma_i}{\sum_{i=1}^r \sigma_i} \geq \tau$
12: Define dominant subspace $\{\mathbf{u}_1, \ldots, \mathbf{u}_k\}, \{\mathbf{v}_1, \ldots, \mathbf{v}_k\}$
13: **Safe Update Construction:**
14: Initialize $\Delta \mathbf{W}_{\text{safe}} \leftarrow 0$
15: **for** $i = 1$ to $m$ **do**
16:     **for** $j = 1$ to $n$ **do**
17:         **if** $i > k$ **and** $j > k$ **then**
18:             $\Delta \mathbf{W}_{\text{safe}} \leftarrow \Delta \mathbf{W}_{\text{safe}} + \alpha_{ij} \mathbf{u}_i \mathbf{v}_j^\top$
19:         **end if**
20:     **end for**
21: **end for**
22: **return** $\Delta \mathbf{W}_{\text{safe}}$

---

# E  EXPERIMENTAL DETAIL

## E.1  BASELINES

- **MEMIT** (Meng et al., 2022b) is the first method to support large-scale knowledge injection across multiple layers. It exploits the key–value structure (Geva et al., 2021) of FFNs and improves upon ROME by restricting updates to a contiguous set of layers, allowing thousands of new facts to be inserted in one pass. However, MEMIT does not consider sequential editing, leaving space for later improvements.

- **RECT** (Gu et al., 2024)RECT is designed to mitigate the degradation of general abilities during sequential editing. It observes that general ability performance declines as more edits are applied, and addresses this by updating parameters based on the magnitude of change in individual weights. However, as our earlier analysis suggests, general abilities are governed by mappings between directions rather than individual parameters. Consequently, RECT remains too **localized** at the parameter level and fails to effectively preserve general abilities in long-horizon sequential editing.

- **PRUNE** (Ma et al., 2024) is specifically designed for sequential editing with the goal of protecting the general abilities of LLMs. From the perspective of matrix conditioning, it constrains the

maximum singular value of the update matrix so that it does not exceed that of the original parameter matrix, thereby reducing the risk of collapse. However, unlike our method, PRUNE does not filter the directions associated with large singular values, which may weaken knowledge retention. Moreover, its constraint only limits singular values to remain below a threshold, effectively attenuating but not eliminating the influence of noise. As a result, PRUNE still struggles to maintain general abilities under long-horizon sequential editing.

- **NSE** (Jiang et al., 2025b) is a method specifically designed for sequential knowledge editing. It preserves the original parameters during update computation, ensuring that each new edit does not interfere with previously injected knowledge. Inspired by the key–value view of FFN layers (Geva et al., 2021), NSE treats each $(k, v)$ pair as a neuron and uses activation values to identify those neurons most relevant to the current update, restricting parameter changes within this subset. While this reduces unnecessary disturbance to the model, neuron-level selection alone cannot fully protect general abilities due to the problem of *superposition*, where a single neuron may encode multiple orthogonal directions. As a result, NSE still fails to maintain general abilities under long-horizon sequential editing.

- **AlphaEdit** (Fang et al., 2024) is a method specifically designed for sequential knowledge editing. It constructs a protection subspace for previously stored knowledge by collecting 100K (subject, relation, object) triples from Wikipedia. During subsequent edits, parameter updates are projected onto the null space of this protection subspace to prevent interference with existing knowledge. However, based on our earlier analysis, sequential editing primarily perturbs the subspace associated with general abilities rather than factual knowledge alone. Thus, the choice of protection subspace in AlphaEdit is not sufficiently precise. As shown in our experiments, AlphaEdit can withstand more editing steps compared to other baselines, but eventually still suffers from a collapse of general abilities.

## E.2 DATASETS

- **ZsRE** Levy et al. (2017) is a question-answering (QA) dataset. Each sample contains a subject string and a corresponding answer, which serve as the editing target to assess **Efficacy**. To evaluate **Paraphrase**, it utilizes rephrased questions generated through back-translation. Following prior work, it employs out-of-scope Natural Questions to measure **Neighborhood** (also referred to as **Locality**).

- **Counterfact** Meng et al. (2022b) is a more challenging dataset that contrasts Counterfactual with factual statements. Each record is derived from an entry in the PARAREL dataset Elazar et al. (2021), with all entities originating from WikiData. It uses metrics similar to ZsRE to evaluate **Efficacy Score**, **Paraphrase Score**, and **Neighborhood Score**. For its out-of-scope data, it replaces the subject entity with an approximate entity that shares the same predicate. Furthermore, Counterfact uniquely includes multiple generation prompts with the same meaning to test the **Fluency**(Generation Entropy) and **Consistency**(Reference Score) of the generated text.

## E.3 ZSRE METRICS

In line with prior work (Meng et al., 2022a;b), we define the evaluation metrics on the ZSRE dataset. Given a language model $f_\theta$, a factual prompt $(s_i, r_i)$, its target output $o_i$, and the model's pre-edit prediction $o_i^c$, the following metrics are used:

- **Efficacy**: This metric reflects the model's accuracy on the edited samples, computed as the average top-1 success rate:
$$\mathbb{E}_i \left\{ o_i = \arg\max_o \mathbb{P}_{f_\theta}(o \mid (s_i, r_i)) \right\}. \tag{7}$$

- **Paraphrase**: This measures how well the model transfers the edit to paraphrased forms of $(s_i, r_i)$, denoted as $N((s_i, r_i))$. It is defined as the average top-1 accuracy over these rephrasings:
$$\mathbb{E}_i \left\{ o_i = \arg\max_o \mathbb{P}_{f_\theta}(o \mid N((s_i, r_i))) \right\}. \tag{8}$$

- **Neighborhood**: This evaluates whether unrelated prompts $O(s_i, r_i)$ remain unaffected by the edit. It is measured as the proportion of cases where predictions for such prompts stay consistent:
$$\mathbb{E}_i \left\{ o_i^c = \arg\max_o \mathbb{P}_{f_\theta}(o \mid O((s_i, r_i))) \right\}. \tag{9}$$

### E.4 COUNTERFACT METRICS

In line with prior work (Meng et al., 2022a;b), we further define the metrics used in the Counterfact benchmark. Given a language model $f_\theta$, a factual prompt $(s_i, r_i)$, a target output $o_i$, and the model's pre-edit prediction $o_i^c$, we define:

- **Efficacy Score**: The fraction of cases where, for the prompt $(s_i, r_i)$, the target $o_i$ receives higher probability than the original output $o_c^i$:

$$\mathbb{E}_i \left[ \mathbb{P}_{f_\theta}[o_i \mid (s_i, r_i)] > \mathbb{P}_{f_\theta}[o_c^i \mid (s_i, r_i)] \right]. \qquad (10)$$

- **Paraphrase Score**: The proportion of paraphrased prompts $N((s_i, r_i))$ where the edited output $o_i$ is ranked higher than the original response $o_i^c$:

$$\mathbb{E}_i \left[ \mathbb{P}_{f_\theta}[o_i \mid N((s_i, r_i))] > \mathbb{P}_{f_\theta}[o_c^i \mid N((s_i, r_i))] \right]. \qquad (11)$$

- **Neighborhood Score**: The proportion of semantically related but distinct prompts $O((s_i, r_i))$ where the model maintains correct predictions, assigning higher probability to $o_i$ over $o_c^i$:

$$\mathbb{E}_i \left[ \mathbb{P}_{f_\theta}[o_i \mid O((s_i, r_i))] > \mathbb{P}_{f_\theta}[o_c^i \mid O((s_i, r_i))] \right]. \qquad (12)$$

- **Fluency**: A measure of output repetition, defined using the entropy of the n-gram distribution:

$$-\frac{2}{3} \sum_k g_2(k) \log_2 g_2(k) + \frac{4}{3} \sum_k g_3(k) \log_2 g_3(k), \qquad (13)$$

where $g_n(\cdot)$ denotes the frequency distribution over $n$-grams.

- **Consistency**: This evaluates how consistent the model's generations are with external references. Given a subject $s$, we compute the cosine similarity between TF-IDF embeddings of the model's text and the corresponding Wikipedia article about $o$.

### E.5 DETAILS OF GLUE

GLUE (Wang et al., 2019) is a comprehensive benchmark, and this paper leverages the following six subtasks:

- **SST** (Socher et al., 2013) (Stanford Sentiment Treebank): A single-sentence classification task based on movie reviews, where the goal is to predict binary sentiment labels.

- **MRPC** (Dolan & Brockett, 2005) (Microsoft Research Paraphrase Corpus): A sentence-pair task aiming to identify whether two sentences are semantically equivalent.

- **MMLU** (Hendrycks et al., 2021) (Massive Multi-task Language Understanding): A broad benchmark covering diverse subjects, designed to evaluate models under zero-shot and few-shot conditions.

- **RTE** (Bentivogli et al., 2009) (Recognizing Textual Entailment): A natural language inference task where the objective is to determine if a premise entails its corresponding hypothesis.

- **CoLA** (Warstadt et al., 2019) (Corpus of Linguistic Acceptability): A single-sentence classification benchmark that tests whether sentences are grammatically acceptable.

- **NLI** (Williams et al., 2018) (Natural Language Inference): A task requiring models to infer the logical relationship between a premise and a hypothesis.

### E.6 METHOD IMPLEMENTATION DETAILS

All experiments based on GPT-J and LLaMA3 are conducted on NVIDIA A800 GPUs with 80GB memory, while experiments involving GPT2-XL are performed on NVIDIA RTX 4090 GPUs with 24GB memory. For baselines, we directly adopt the official implementations of ALPHAEDIT and NSE without modifying their original hyperparameter configurations. The only additional hyperparameter introduced by our method is the singular value projection threshold. Following the results in Appendix F.5, we consistently set this threshold to preserve the top 10% singular values across all models. This choice is justified as our projection strategy is independent of the specific baseline but only depends on the underlying model.

### E.7 RELATED WORK (FULL VERSION)

**Parameter-Preserving Methods.** Parameter-preserving approaches maintain the base model's parameters unchanged and instead incorporate external mechanisms to realize edits. A common direction is to attach additional modules that store and retrieve edited knowledge. For example, SERAC (Mitchell et al., 2022b) introduces an auxiliary memory with a Counterfactual model, CaliNet (Dong et al., 2022) and T-Patcher (Huang et al., 2023) insert neuron-based units, and GRACE (Hartvigsen et al., 2023) organizes edits in a dynamic codebook. MELO (Yu et al., 2024) uses additional LoRA-style adapters to preserve original parameters, while WISE (Wang et al., 2024a) improves stability and general ability with dual-memory and conflict-free sharding. Another line of work performs edits through prompting: MemPrompt (Madaan et al., 2022) and IKE (Zheng et al., 2023) rely on injecting new facts into the input context. More recent efforts combine symbolic structures with neural editing, such as OneEdit (Zhang et al., 2024c), which leverages knowledge graphs for collaborative knowledge updates.

**Parameter-Modifying Methods.** Parameter-modifying methods directly update the model's weights to encode new knowledge. Meta-learning based techniques predict parameter shifts through hypernetworks, including MEND (Mitchell et al., 2022a), MALMEN (Tan et al., 2024), and InstructEdit (Zhang et al., 2024b). Locate-then-edit methods first determine the locations where knowledge is stored and then apply targeted modifications. Typical examples are ROME (Meng et al., 2022a), which computes updates using closed-form equations, MEMIT (Meng et al., 2022b), which scales editing to batches, GLAME (Zhang et al., 2024a), which integrates knowledge graphs, and AnyEdit (Jiang et al., 2025a), which recursively edits knowledge of arbitrary structure. When edits are carried out in a sequential manner, however, additional difficulties arise. Consecutive updates can accumulate interference and eventually harm model performance. To counter these issues, several improvements have been proposed: RECT (Gu et al., 2024) enforces sparsity on update parameters at single parameter level, PRUNE (Ma et al., 2024) controls the condition number of parameter updates, AlphaEdit (Fang et al., 2024) constrains modifications to a null space of previous stored knowledge, and NSE (Jiang et al., 2025b) select the modification position that contributes the most to knowledge storage based on the activation values of the neurons.

## F DETAILED EXPERIMENT RESULTS

### F.1 ROBUSTNESS UNDER OUTPUT-SIDE PERTURBATIONS ACROSS SINGULAR VALUE GROUPS.

**Symmetric Output-side Perturbation.** To complement the input-side analysis in the main text, we also evaluate robustness under *output-side perturbations*. Specifically, for a chosen group of left singular vectors $\mathcal{H}$ (partitioned by cumulative energy of singular values in the same way as before), we inject structured rank-one perturbations of the form:

$$\mathbf{\Delta} = \sum_{i \in \mathcal{H}} \sum_{j=1}^{r} \beta_{i,j} \, \mathbf{u}_i \mathbf{v}_j^\top, \qquad \beta_{i,j} \sim \mathcal{N}(0, 1).$$

The perturbation is normalized and scaled to fixed strength as

$$\tilde{\mathbf{\Delta}} = \varepsilon \cdot \frac{\mathbf{\Delta}}{\|\mathbf{\Delta}\|_F}.$$

The resulting perturbed weight matrix is $\mathbf{W}' = \mathbf{W} + \tilde{\mathbf{\Delta}}$, which can be interpreted as altering the *input representation* of selected outputs $\{\mathbf{u}_i\}_{i \in \mathcal{H}}$ (left singular vector) into random mixtures of all inputs $\{\mathbf{v}_j\}_{j=1}^{r}$ (right singular vector). We report the corresponding robustness curves across different output groups in Figure 11, and the observed trends are consistent with the input-side perturbation experiments.

### F.2 A SPECTRUM ANALYSIS COMPARISO BETWEEN ALPHAEDIT AND MEMIT-REVIVE

In Section 2.3, we present the SS performance of MEMIT under long editing sequences, where we observe that significant shifts in critical subspace singular vectors emerge after around twenty

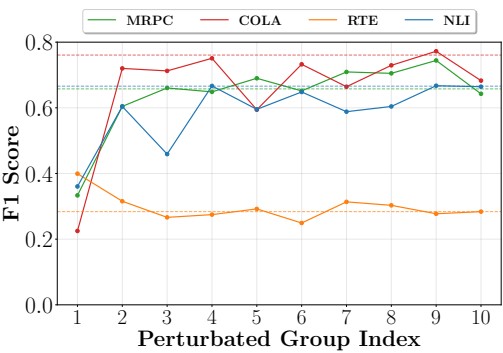

Figure 11: Sensitivity of general ability to perturbations across different output side (left singular vector) spectral groups.

rounds of editing. In this section, we analyze and record the SS dynamics of the current strongest baseline, ALPHAEDIT, under long-sequence editing, and compare them with MEMIT-REVIVE. The experimental setup involves editing 10,000 samples (100 samples per round for 100 rounds) from the COUNTERFACT dataset on the LLAMA3 model. As illustrated in Figure 12, ALPHAEDIT maintains relatively small shifts in the critical subspace vectors during the early editing rounds, but its maximum SS inevitably decreases as editing proceeds. By the end of the editing process, the maximum SS drops below 0.3, which aligns with its performance degradation on the GLUE benchmark in Figure 7. In contrast, MEMIT-REVIVE consistently preserves an SS maximum value of 1 throughout the entire editing sequence (as illustrated in Figure 13), indicating the stability of its critical vector subspace, which also corresponds well with its stable performance on GLUE. Overall, these results demonstrate that our REVIVE method effectively safeguards the critical vector subspace, ensuring that the model's general capabilities remain stable under long-sequence editing.

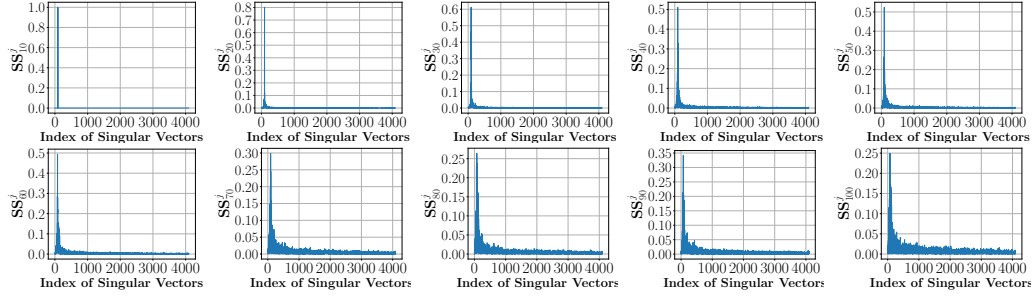

Figure 12: Evolution of SS of AlphaEdit over sequential editing, from $SS_{10}$ to $SS_{100}$ with step size 10.

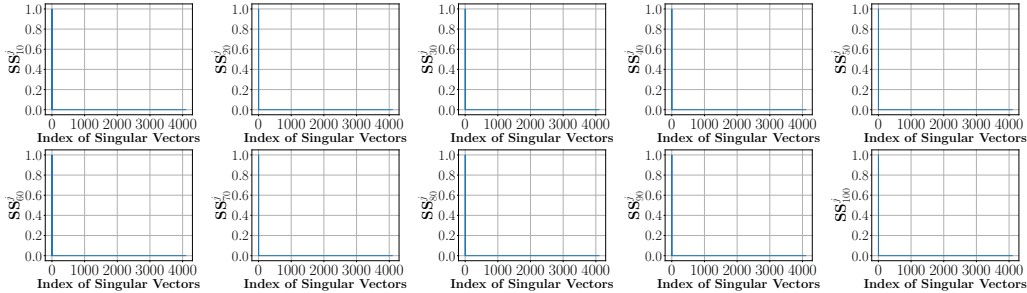

Figure 13: Evolution of SS of MEMIT-REVIVE over sequential editing, from $SS_{10}$ to $SS_{100}$ with step size 10.

### F.3 A FINE-GRAINED ANALYSIS ON LEFT VECTORS CHANGES

In the main text, we report the variations of the right singular vectors, while here we illustrate the changes of the left singular vectors. It can be observed from Figure 14 that both exhibit essentially the same trend.

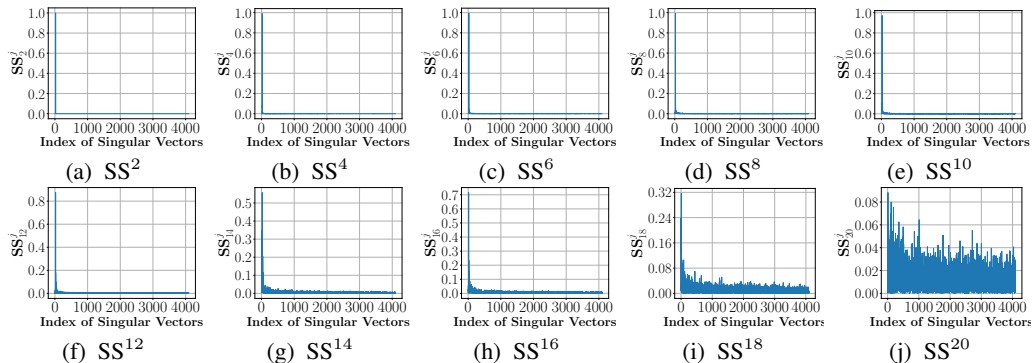

(a) $SS^2$   (b) $SS^4$   (c) $SS^6$   (d) $SS^8$   (e) $SS^{10}$

(f) $SS^{12}$   (g) $SS^{14}$   (h) $SS^{16}$   (i) $SS^{18}$   (j) $SS^{20}$

Figure 14: Evolution of Left Singular Vector Similarity (SS) over sequential editing.from $SS_2$ to $SS_{20}$ with step size 2.

### F.4 FULL GLUE RESULTS

Here, we present the remaining GLUE evaluation metrics that were omitted in Section 4.2 due to space constraints, The results of the rest three datasets are presented in Figure 15.

### F.5 FULL THRESHOLD EXPERIMENTS RESULTS

Since the projection threshold is primarily related to the model itself and less influenced by the chosen method, we only present the performance variation of MEMIT-REVIVE across three different models with respect to the projection threshold. The detailed changes in editing metrics with varying thresholds are recorded in Table 2, while Figure 16 shows the variation in the LLaMA3 model's performance on the GLUE benchmark with different projection thresholds. Note that the GPT-J and GPT2-XL models, due to their relatively poor performance on GLUE even before editing, are not included in the results presented here.

### F.6 ANALYSIS ON BATCHSIZE

Here, we investigate the model's robustness to the edit batch size. We conduct an experiment with a fixed total of 5,000 editing samples from COUNTERFACT, while varying the batch size per edit as 100, 200, and 500. As shown in Figure 17, prior baselines are highly sensitive to changes in batch size. This observation supports our earlier hypothesis: as the number of edits increases, perturbations along specific input–output directions accumulate, leading to model collapse. In contrast, when our REVIVE is integrated, the baselines exhibit stable performance that does not fluctuate significantly with batch size.

### F.7 REVIVE ENHANCED BASELINES UNDER EXTREME SETTINGS

Here, we further present the complete results of the REVIVE method under extreme test conditions, including its performance on the ZsRE dataset, which was not shown in the main text.

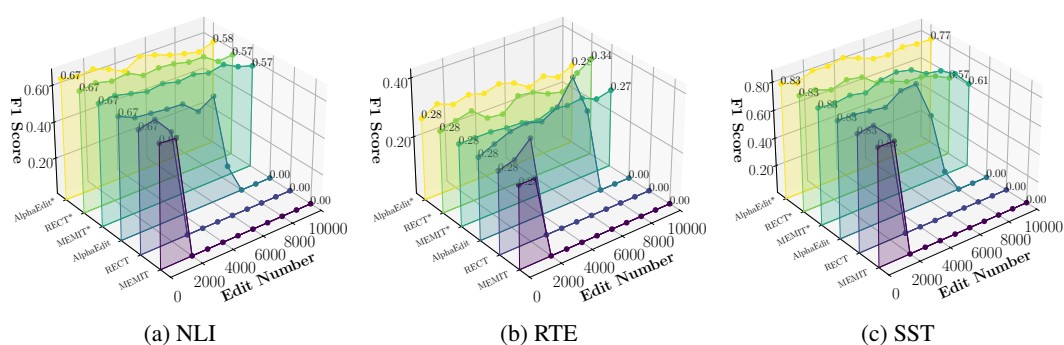

Figure 15: Baseline and corresponding REVIVE version(*) performance on GLUE across datasets.

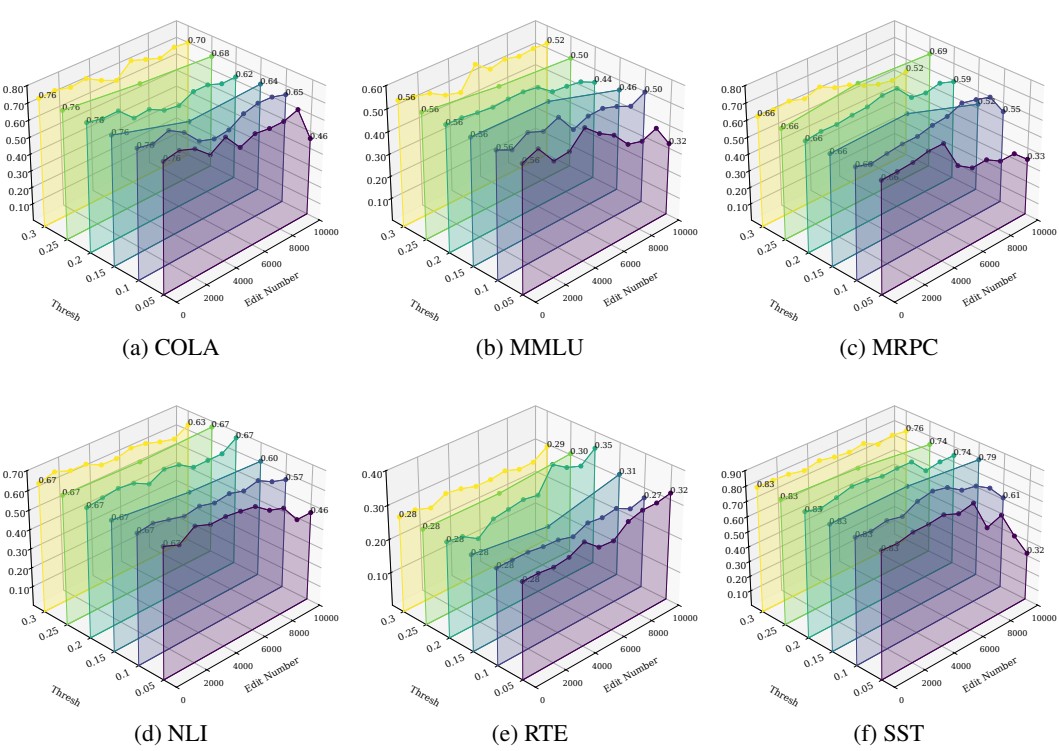

Figure 16: GLUE evaluation results on LLaMA3 after 10,000 edits on the CounterFact dataset using MEMIT-REVIVE with different protection thresholds.

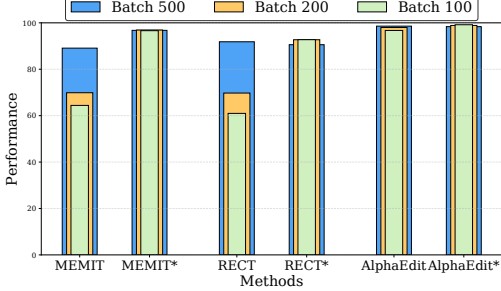

Figure 17: Performance of methods and their REVIVE-enhanced versions under different batch sizes. * denotes methods with REVIVE.

Table 2: Performance results of MEMIT-REVIVE on sequential editing task under different singular value energy thresholds (10,000 Samples from CounterFact).

| Model | Thresh | Counterfact | | | | | ZsRE | | |
|---|---|---|---|---|---|---|---|---|---|
| | | Eff.↑ | Para.↑ | Neigh.↑ | Flu.↑ | Consis.↑ | Eff.↑ | Para.↑ | Neigh.↑ |
| LLaMA3 | 0.05 | 94.46 | 86.03 | 59.70 | 587.30 | 29.83 | 78.66 | 75.89 | 29.29 |
| | 0.10 | 95.62 | 84.60 | 62.17 | 603.22 | 29.39 | 86.56 | 83.07 | 31.88 |
| | 0.15 | 95.03 | 80.60 | 64.49 | 613.66 | 29.19 | 87.10 | 83.36 | 32.41 |
| | 0.20 | 94.58 | 78.38 | 66.19 | 621.15 | 29.63 | 86.85 | 83.46 | 32.75 |
| | 0.25 | 92.96 | 73.94 | 68.94 | 624.63 | 29.49 | 83.85 | 80.23 | 33.27 |
| | 0.30 | 88.94 | 67.56 | 71.86 | 625.68 | 28.84 | 81.18 | 77.81 | 33.02 |
| GPT-J | 0.05 | 91.23 | 83.72 | 57.26 | 596.20 | 33.29 | 78.50 | 73.19 | 27.44 |
| | 0.10 | 97.09 | 87.01 | 67.10 | 616.15 | 40.00 | 83.87 | 77.28 | 29.77 |
| | 0.15 | 96.74 | 81.20 | 69.98 | 617.42 | 39.63 | 88.57 | 82.87 | 29.15 |
| | 0.20 | 94.95 | 76.59 | 72.13 | 621.01 | 38.36 | 85.83 | 79.97 | 29.27 |
| | 0.25 | 92.84 | 69.42 | 74.15 | 621.61 | 37.19 | 81.67 | 74.67 | 27.59 |
| | 0.30 | 88.49 | 64.30 | 74.94 | 623.53 | 36.66 | 81.32 | 73.54 | 28.56 |
| GPT2-XL | 0.05 | 91.89 | 80.72 | 61.13 | 575.14 | 32.12 | 62.13 | 55.40 | 25.90 |
| | 0.10 | 90.82 | 77.24 | 63.73 | 595.36 | 34.28 | 63.34 | 55.29 | 25.93 |
| | 0.15 | 87.82 | 73.39 | 65.89 | 607.06 | 35.33 | 66.19 | 58.40 | 27.13 |
| | 0.20 | 83.10 | 66.95 | 68.44 | 615.17 | 35.46 | 64.53 | 57.45 | 26.60 |
| | 0.25 | 78.77 | 61.82 | 69.66 | 618.28 | 35.17 | 58.11 | 51.80 | 26.89 |
| | 0.30 | 73.03 | 57.28 | 71.12 | 621.46 | 34.60 | 57.05 | 51.15 | 26.42 |

Table 3: Performance results of REVIVE enhanced Baseilnes under extreme sequential editing (20000 edits).

| Model | Method | Counterfact | | | | | ZsRE | | |
|---|---|---|---|---|---|---|---|---|---|
| | | Eff.↑ | Para.↑ | Neigh.↑ | Flu.↑ | Consis.↑ | Eff.↑ | Para.↑ | Neigh.↑ |
| LLaMA3 | MEMIT-REVIVE | 91.94 | 79.67 | 56.90 | 557.61 | 26.44 | 84.11 | 79.85 | 32.92 |
| | RECT-REVIVE | 89.00 | 76.78 | 60.54 | 594.38 | 27.93 | 79.35 | 76.35 | 30.24 |
| | AlphaEdit-REVIVE | 97.50 | 87.24 | 57.65 | 613.22 | 32.77 | 92.62 | 88.25 | 31.31 |
| | NSE-REVIVE | 98.50 | 90.38 | 61.78 | 615.65 | 33.23 | 93.91 | 89.67 | 31.58 |

