# OpenReview forum: "ReviveEdit: Robust Sequential Editing via Dominant Subspace Preservation"
_ICLR.cc/2026/Conference — ICLR 2026 Conference Withdrawn Submission_

### Official Review · Reviewer_qf6z · 2025-10-17

**Soundness:** 3
**Presentation:** 2
**Contribution:** 2
**Rating:** 2
**Confidence:** 5

**Summary:**

This paper investigates the problem of performance degradation in large language models (LLMs) during sequential knowledge editing (SME).
The authors perform a singular value decomposition (SVD) on the model’s parameter matrices and observe that a model’s general capabilities are largely concentrated within its dominant singular subspace. They argue that continuous edits progressively distort this subspace, ultimately leading to model collapse.
To address this issue, the paper proposes ReviveEdit, a method that, during each edit, performs projection and filtering operations to constrain updates only to the low-energy directions of the parameter space. This procedure preserves the integrity of the dominant singular subspace and prevents degradation of general abilities.
Experiments on GPT-J, LLaMA3, and other models, using datasets such as COUNTERFACT, ZSRE, and GLUE, demonstrate that ReviveEdit achieves higher editing success rates and stronger stability compared to prior methods. Notably, even after 20,000 sequential edits, the model retains approximately 86% of its downstream task performance.

**Strengths:**

**Strengths:**

The experiments are **comprehensive and well-designed**.
They cover multiple models and tasks, including long-horizon sequential editing, and the results are stable and clearly reported. This demonstrates that the authors have invested substantial effort in the experimental evaluation.

The work also has **clear engineering value**.
The proposed method is simple, modular, and can be easily integrated with other model editing frameworks, making it practically useful for improving model robustness in large-scale applications.

**Weaknesses:**

**Weakness:**

The core idea of this paper — preserving dominant directions in the parameter matrix’s feature or singular subspace — is not novel.
Earlier works such as Delta-Edit (2024) and O-Edit (2025) have already proposed highly similar motivations from different perspectives, namely low-rank perturbation (Delta Projection) and orthogonal subspace regularization (gradient-space orthogonality).

The notion of “protecting the dominant subspace” has also been well established in prior research, including studies on low-rank fine-tuning, model compression, and weight perturbation analysis.
As a result, this paper’s theoretical contribution is limited, representing more of a formal restatement or spectral reinterpretation of existing ideas rather than a genuinely new conceptual advance.

**Therefore, despite the solid empirical validation, this work does not meet the originality threshold expected for ICLR acceptance.**

[1] O-EDIT: ORTHOGONAL SUBSPACE EDITING FOR LAN GUAGE MODEL SEQUENTIAL EDITING

[2] DeltaEdit: Enhancing Sequential Editing in Large Language Models by Controlling Superimposed Noise

**Questions:**

See **Weankess**

---

> ### Author Response · Authors · 2025-11-26
> **Reply（part 1/1）**
>
> **(W1):The core idea of this paper — preserving dominant directions in the parameter matrix’s feature or singular subspace — is not novel. Earlier works such as Delta-Edit (2024) and O-Edit (2025) have already proposed highly similar motivations from different perspectives, namely low-rank perturbation (Delta Projection) and orthogonal subspace regularization (gradient-space orthogonality).**
>
> We understand why the reviewer draws a connection, as all three methods involve constraining updates within certain “subspaces.” However, the problem setting, the source of protected subspaces, and the theoretical assumptions differ fundamentally, so the core ideas do not overlap. Specifically, Delta-Edit (2024) and O-Edit (2025) focus on **inter-edit interference**: they track directions of previous edits and project new updates to avoid overwriting past changes. Their protected subspace comes from **accumulated edit history**, addressing the question of how to prevent new edits from conflicting with previous ones.&#x20;
>
> In contrast, our REVIVE targets a different failure mode (Section 2) : **progressive corruption of the dominant singular subspace of FFN parameters** (energy decay and directional rotation), which arises from the **model’s intrinsic spectral structure**, and accumulates even when edits are unrelated. REVIVE thus preserves the dominant singular directions of the parameter matrix, crucial for general model capabilities, rather than historical edit directions.&#x20;
>
> **While all methods use projection, Delta-/O-Edit handles interference between edits, whereas REVIVE targets a fundamentally different and previously unrecognized issue: model-level degradation of the dominant singular subspace during long editing sequences. These are two entirely different perspectives on sequential editing,&#x20;**&#x61;nd REVIVE addresses a failure mode that prior work has not identified. This distinction makes the core idea of our approach novel.
>
>
>
> **(W2):The notion of “protecting the dominant subspace” has also been well established in prior research, including studies on low-rank fine-tuning, model compression, and weight perturbation analysis. As a result, this paper’s theoretical contribution is limited, representing more of a formal restatement or spectral reinterpretation of existing ideas rather than a genuinely new conceptual advance.**
>
> Our main contribution is the first to link real-world *sequential knowledge editing* with spectral changes in FFN parameter matrices, **revealing why long-horizon edits fail**. Prior work has discussed weight perturbations, low-rank updates, or subspace constraints, but none explains **why existing methods inevitably degrade after a few thousand edits or identifies the underlying mechanism**. Because our analysis uncovers a different cause of collapse in sequential editing, the subspace we protect is fundamentally different as a direct consequence.
>
> By tracking the singular spectrum of FFN weights, we show that the dominant singular subspace erodes over sequential edits: high-energy directions rotate and drift, directly impacting general model abilities. Targeted interventions confirm the causal chain: &#x20;
>
> **Disruption of the dominant singular subspace → degradation of general abilities → sequential editing failure.**
>
> Building on this insight, we propose REVIVE, a lightweight, plug-and-play mechanism that preserves the dominant singular subspace, filters only harmful update components, and prevents cumulative spectral degradation. **Experiments demonstrate that while prior methods degrade after 2k–8k edits**, **REVIVE enables stability beyond 20k edits**, maintaining both high editing success and general capabilities.&#x20;
>
> **In short, REVIVE makes large-scale, long-term sequential knowledge editing feasible and introduces a spectral perspective and mechanism-level foundation for future work.**
>
> &#x20;

---

> ### Comment · Reviewer_qf6z · 2025-11-27
>
> I have decided to raise my score to 4.

---

### Official Review · Reviewer_CBPd · 2025-10-26

**Soundness:** 2
**Presentation:** 2
**Contribution:** 1
**Rating:** 2
**Confidence:** 3

**Summary:**

This paper proposes REVIVEEDIT, a framework for robust sequential model editing that prevents model collapse by preserving the dominant singular subspace of parameter matrices. Through spectral analysis, the authors argue that catastrophic degradation during sequential edits stems from the corruption of high-energy singular components that encode general abilities. REVIVEEDIT mitigates this by projecting updates onto the singular vector basis and removing directions that interfere with dominant subspaces.

Experiments across GPT2-XL, GPT-J, and LLaMA3 on COUNTERFACT and ZSRE show substantial gains in both editing efficacy and general ability preservation, even after tens of thousands of sequential edits.

**Strengths:**

1.	The paper provides an insightful spectral explanation of why sequential editing leads to degradation, connecting weight structure and general ability loss.
2.	REVIVEEDIT is plug-and-play and compatible with existing editing frameworks such as MEMIT and AlphaEdit.
3.	Strong empirical validation across multiple models and baselines, including large-scale tests.

**Weaknesses:**

1.	Limited novelty compared to prior work –The method’s core idea—preserving or constraining updates within structured low-rank subspaces—bears strong resemblance to previous works such as PRUNE and AlphaEdit, which also regulate parameter updates through rank or null-space constraints. The new contribution (dominant subspace preservation via SVD) can be seen as an incremental extension of these ideas rather than a fundamentally new paradigm.
2.	The use of SVD projection and component filtering is technically straightforward. The novelty mainly lies in the empirical finding that high-singular-value directions encode general abilities, but this is somewhat intuitive and overlaps with insights from AlphaEdit.
3.	Computational feasibility not fully addressed. Performing SVD for all large matrices is costly; no discussion of efficiency or scalability to very large LLMs (e.g., 70B parameters) is provided.
4.	The argument that dominant subspace corruption is the cause of collapse remains empirical; a stronger theoretical guarantee or causal analysis is missing.

**Questions:**

1.	Does REVIVEEDIT require recomputing SVD after each batch of edits, or is it fixed once? How does that affect computational efficiency?

---

> ### Author Response · Authors · 2025-11-26
> **Reply（part1/2）**
>
> **(W1 + W2) The paper’s core idea—preserving updates within structured low-rank or dominant subspaces—appears conceptually similar to prior work such as PRUNE and AlphaEdit, both of which also regulate parameter updates through rank or null-space constraints. The use of SVD-based projection and component filtering is technically straightforward, and the claimed insight that high-singular-value directions encode general abilities seems intuitive and overlaps with observations made in AlphaEdit. Overall, the contribution may be incremental rather than representing a fundamentally new paradigm.**
>
> We appreciate the reviewer’s comparison, and we would like to clarify that the similarity in form (i.e., both methods using projection) does not imply similarity in mechanism or insight. **The key differences lie in what failure mode each method targets and which subspace each method chooses to protect.**
>
> First, although AlphaEdit also employs projection, it operates on a fundamentally different premise. AlphaEdit is **data-driven and extrinsic**: it constructs a protection subspace from input activations of **100k sampled factual triples** to mitigate knowledge interference. In contrast, REVIVE is **model-driven and intrinsic**: our spectral analysis shows that long-sequence degradation comes from **corruption of the dominant singular subspace** of the weight matrices themselves. Thus, rather than relying on additional data, we explicitly protect the **model’s own functional operator structure**. This distinction explains the empirical divergence: AlphaEdit begins to deteriorate around 8k edits, whereas methods augmented with REVIVE remain stable beyond **20k edits**.
>
> Second, **PRUNE suppresses any update singular value** that exceeds the largest singular value of the original parameter matrix, **without considering** the directions associated with those singular values. This magnitude-only suppressio&#x6E;**&#x20;ignores** which components are actually harmful or beneficial: it may discard useful update directions while weakening the edit itself. As a result, **PRUNE cannot preserve the model’s functional subspace** and performs poorly in long sequential editing. In contrast, **REVIVE explicitly preserves these directions** by filtering out only the components that would distort them—addressing a degradation mode that PRUNE is not designed to handle.
>
> Finally, our novelty does not lie in using SVD or projection—these are standard tools—but in **identifying a previously overlooked failure mode** and choosing the protected subspace based on the model’s spectral structure, rather than on external knowledge statistics or uniform magnitude suppression. **This principled subspace selection is what enables stable editing at scales that prior methods cannot sustain.**

---

> ### Author Response · Authors · 2025-11-26
> **Reply（part2/2）**
>
> **(W3 + Q1) The computational feasibility of the method remains unclear. Performing SVD on large weight matrices can be costly, and the paper does not sufficiently discuss the efficiency or scalability of REVIVEEdit when applied to very large LLMs (e.g., 70B parameters). In addition, it is unclear whether REVIVEEdit requires recomputing the SVD after each batch of edits or relies on a fixed decomposition, and how this choice affects overall computational efficiency.**
>
> We thank the reviewer for raising this point. For question 1, We confirm that REVIVEEdit does recompute the SVD for the relevant matrices at each new edit. **Given this, it is natural to consider the computational cost of performing SVD repeatedly, so we provide a detailed analysis below to answer weakness 3.**
>
> It is worth emphasizing that mainstream knowledge-editing methods **do not update the full model**. Instead, they apply highly localized edits to a **small subset** of FFN down-projection matrices, which prior work has consistently identified as the modules where edits actually take effect. For instance, in LLaMA3-8B, typically **only 5 out of 32** `down_proj` matrices are modified. REVIVE follows this standard setup, so **SVD** is required **only for these 5 matrices**.
>
> In our environment, performing `torch.linalg.svd` on a single LLaMA3-8B `down_proj` matrix takes about **2.15 seconds**. This means each full REVIVE update adds only approximately 10 seconds of overhead. Even over 10,000 sequential editing , this amounts to just **\~16.6 minutes of extra time**.
>
> Furthermore, we evaluated the SVD cost for larger models not yet explored by existing knowledge-editing work. For LLaMA3-70B, whose `down_proj` matrices are (28672, 8192), a single SVD takes only **10.2** seconds. This implies that editing 10,000 facts would require an additional \~85 minutes under the same setup.
>
> Despite this modest cost, REVIVE delivers **over 50% improvement in both editing and GLUE performance**, and substantially enhancing stability in long-horizon sequential editing. From a cost–benefit perspective, the **SVD overhead is very small relative to the gains** and remains fully practical for modern large models.

---

### Official Review · Reviewer_x4j5 · 2025-10-27

**Soundness:** 2
**Presentation:** 3
**Contribution:** 2
**Rating:** 4
**Confidence:** 4

**Summary:**

This paper considers sequential knowledge editing in LLMs and argues that catastrophic degradation after long edit sequences arises from corruption of the dominant singular subspace of weight matrices. The authors propose REVIVE, a plug-and-play mechanism that i) decomposes each update $\Delta W$ in the SVD basis of the original weight $W$ and ii) filters out components that involve the top singular directions using an energy threshold $\tau$. Experiments demonstrate substantial gains across multiple editors and model families, while maintaining robustness for sequences of edits up to 20k.

**Strengths:**

1. The paper provides a coherent explanation that deterioration of general abilities is linked to the dominant singular subspace.

2. REVIVE can be applied to existing editors directly, while acknowledging some added compute and storage for SVD-based filtering.

3. The authors conducted extensive experiments to validate the effectiveness of the proposed method.

**Weaknesses:**

1. The core idea—preserving a knowledge subspace—is conceptually close to strands in continual learning and to AlphaEdit in model editing (though AlphaEdit emphasizes feature subspaces while REVIVE emphasizes parameter subspaces).

2. Post-hoc projection may be suboptimal. If the edit intrinsically lies within the top singular subspace, filtering may prevent achieving the desired update.  It would be more principled to include constraint in the solution (similar to AlphaEdit) or even in the optimization (i.e., in finding the target output).

**Questions:**

1. Are there concrete cases where the desired edit demonstrably lies in the top-energy directions?
2. With thousands of edits, does the projected subspace become saturated?
3. Some parameter-preserving baselines should be considered. Recent SimIE is also plug-and-play and reports strong performance, and should be considered as a baseline.
4. Fig. 4 suggests that editing performance decreases suddenly, while low-rank subspace similarity decays more gradually (Fig. 3). What mechanism explains this discrepancy?
5. Could you add an ablation that solves the edit with the constraint (e.g., constrained LS / projected gradient) and compares it to post-hoc projection?

[1]. Aging with grace: Lifelong model editing with discrete key-value adaptors, NeurIPS 2023.

[2]. Towards lifelong model editing via simulating ideal editor, ICML 2025.

---

> ### Author Response · Authors · 2025-11-26
> **Reply（part1/3）**
>
> **(W1) The core idea—preserving a knowledge subspace—is conceptually close to strands in continual learning and to AlphaEdit in model editing (though AlphaEdit emphasizes feature subspaces while REVIVE emphasizes parameter subspaces).**
>
> We thank the reviewer for this insightful connection. While REVIVE shares the broad mathematical concept of subspace projection with CL and AlphaEdit, we emphasize that our core contribution lies in identifying the specific *mechanism&#x20;*&#x6F;f model collapse, which dictates a fundamentally different subspace definition.
>
> * Unlike prior works that apply heuristic constraints, our method is grounded in a novel spectral analysis (Section 2). Through extensive experiments, we empirically discovered that the catastrophic degradation of general abilities is structurally rooted in the **corruption of the dominant singular subspace** of weight matrices. This finding is non-trivial and serves as the primary motivation for our work.
>
> * Based on this insight, REVIVE is designed to explicitly safeguard this specific *intrinsic* structure. This leads to a crucial distinction from AlphaEdit:
>
> **AlphaEdit (Extrinsic/Data-Driven):** It focuses on the **Feature Subspace**. It relies on a retained dataset to define a protection scope based on input activations. It operates on the assumption that preserving activation patterns on sample data preserves capability.
>
> **REVIVE (Intrinsic/Model-Driven):** It focuses on the **Parameter Subspace**. Guided by our discovery, we protect the weight matrix's functional operators directly (via SVD), independent of external data.
>
> **Conclusion:** Therefore, REVIVE mitigates the **underlying structural mechanism** of collapse (spectral corruption) identified in our analysis, whereas AlphaEdit constrains the **consequential representation drift** (activation drift). Our results (e.g., Table1 , Fig. 7 and Fig. 9) confirm that protecting the parameter subspace is more robust for long-term editing than constraining feature subspaces.
>
>
> **(W2) Post-hoc projection may be suboptimal. If the edit intrinsically lies within the top singular subspace, filtering may prevent achieving the desired update. It would be more principled to include constraint in the solution (similar to AlphaEdit) or even in the optimization (i.e., in finding the target output).**
>
> **(Q5) Could you add an ablation that solves the edit with the constraint (e.g., constrained LS / projected gradient) and compares it to post-hoc projection?&#x20;**
>
> We appreciate the reviewer’s question, and we would like to clarify the design motivation behind our approach. REVIVE is intentionally built as a **plug-and-play module** that can be attached to existing knowledge-editing methods **without requiring any modification to their internal optimization procedures**. This modularity is essential for keeping REVIVE simple, general, and widely applicable.
> A natural question is why we apply our projection after the analytic update rather than embedding it directly inside the solver. The key reason is that our projection relies on two-sided spectral information from the parameter matrix, whereas methods like AlphaEdit use a single-sided linear projection that integrates cleanly into their specific solver. Embedding our projection inside the analytic step would alter the structure of the optimization, forcing each editing method to require a new, custom derivation of its solver. This would **break** the plug-and-play nature of REVIVE and make it incompatible with existing frameworks.
> By applying the projection post-solution, we retain two important advantages:
> \- **Full compatibility**: REVIVE works directly with the outputs of all existing analytic editing algorithms. &#x20;
> \- **Solver independence**: No editing method needs to be modified or re-engineered to use REVIVE.
> At the same time, **extensive experiments** demonstrate that this simple post-hoc projection consistently improves stability, locality, and long-horizon performance, confirming the effectiveness of our approach in practice. We will continue to refine and optimize the projection step in future work while preserving its modular, solver-agnostic design philosophy.

---

> ### Author Response · Authors · 2025-11-26
> **Reply（part2/3）**
>
> **(Q1):Are there concrete cases where the desired edit demonstrably lies in the top-energy directions?**
>
>
> We thank the reviewer for the thoughtful question. We examined the failure cases where REVIVE-enhanced methods make mistakes and compared them with the failures of the original baselines. Our statistical analysis did not reveal any systematic pattern suggesting that unsuccessful edits tend to fall into the top-energy directions. We plan to investigate this question more deeply in future work.
>
> It is also worth noting that prior studies have analyzed the semantic roles of singular vectors. For example, previous studies [1][2] show—by projecting singular vectors directly onto the vocabulary—that tail singular vectors correspond to **interpretable lexical or factual patterns**, whereas top singular vectors mainly map to **generic or non-semantic tokens**. This further suggests that top-energy directions are unlikely to encode specific factual edits.
>
> Empirical evidence further supports this: as shown in Table 1, REVIVE achieves near-perfect editing efficacy **(e.g., >95% on CounterFact and ZSRE)** despite explicitly filtering out the dominant singular subspace. If desired factual edits frequently lay in these top-energy directions, our filtering would inherently block them and lead to low efficacy. The **consistently high success rates** therefore imply that factual updates predominantly reside in the low-rank, low-energy subspace, orthogonal to the protected dominant directions.
>
> [1] Sid Black, *The Singular Value Decompositions of Transformer Weight Matrices Are Highly Interpretable*, 2023.
>
> [2] Staats M, Thamm M, Rosenow B. *Small Singular Values Matter: A Random Matrix Analysis of Transformer Models*. NeurIPS, 2025.
>
> **(Q2):With thousands of edits, does the projected subspace become saturated?**
>
> &#x20;In our method, the protected subspace is defined by the cumulative **singular value** ratio, and because the spectrum is strongly long-tailed, the top 10% of singular *values* correspond to only about **3%** of the **singular** **vectors** in LLaMA3-8B. This asymmetry means that roughly **97% of the vectors** remain available for updates, so thousands of edits do not come close to saturating the editable space.  Furthermore, even under extreme sequences with **20k** edits, the edit success rate stays above **93%**(Table 3), indicating that the model’s representational capacity is much larger than we might expect.

---

> ### Author Response · Authors · 2025-11-26
> **Reply（part3/3）**
>
> **(Q3): Some parameter-preserving baselines should be considered. Recent SimIE is also plug-and-play and reports strong performance, and should be considered as a baseline.**
>
> Following the reviewer’s suggestion, we reproduced the results of SimIE. We strictly followed the official implementation of SimIE and conducted a comparative experiment on ZSRE (1,000 sequential edits) using LLaMA3-8B and GPT2-XL. The results are summarized below.&#x20;
>
> Both of the two plug-in modules show strong enhancement of the original baseline. In our experiments, **REVIVE** achieves significantly better performance on both **Eff and Para**, while **SimIE** shows an advantage on **Neigh**.
>
> This difference can be understood from the design goals of the two methods. Because REVIVE preserves the model’s **general abilities** over long edit sequences, it maintains **editability far more effectively** than SimIE, leading to its stronger results on **Eff and Para**. In contrast, SimIE maintains a separate memory matrix for storing past edits , which helps it capture **local edit behavior more precisely** and thus perform better on **Neigh**.
>
> However, we also noticed a practical limitation: SimIE’s cache matrix grows linearly with the number of edits, which means that for very long edit sequences, the method may face scalability issues due to increasing memory and computation costs.
>
> |                  | Llama-3 (8B) |       |        |   | GPT2-XL(1.5B) |       |        |   |
> | ---------------- | ------------ | ----- | ------ | - | ------------- | ----- | ------ | - |
> |                  | Eff.         | Para. | Neigh. |   | Eff.          | Para. | Neigh. |   |
> | Unedited Model   | 0.35         | 0.35  | 0.31   |   | 0.22          | 0.21  | 0.24   |   |
> | MEMIT            | 0.00         | 0.00  | 0.00   |   | 0.58          | 0.50  | 0.13   |   |
> | MEMIT+SIMIE      | 0.61         | 0.48  | 0.22   |   | 0.81          | 0.66  | 0.19   |   |
> | MEMIT+REVIVE     | 0.94         | 0.91  | 0.15   |   | 0.84          | 0.76  | 0.15   |   |
> | AlphaEdit        | 0.86         | 0.78  | 0.29   |   | 0.83          | 0.70  | 0.18   |   |
> | AlphaEdit+SIMIE  | 0.87         | 0.77  | 0.32   |   | 0.88          | 0.76  | 0.20   |   |
> | AlphaEdit+REVIVE | 0.95         | 0.92  | 0.31   |   | 0.95          | 0.89  | 0.18   |   |
>
> **(Q4):Fig. 4 suggests that editing performance decreases suddenly, while low-rank subspace similarity decays more gradually (Fig. 3). What mechanism explains this discrepancy?**
>
> We thank the reviewer for the thoughtful question. We note that there may be a slight mismatch in the figure references in the question, so we first clarify the trends across all relevant plots. In Fig. 3(a), we plot the change in editing accuracy over the editing sequence, while Fig. 3(b) shows the evolution of general capabilities. Fig. 4 reports the similarity of the low-rank subspace as editing progresses.
>
> **The discrepancy comes from the different granularities of the two measurements:** knowledge-level performance reacts to fine-grained parameter changes, whereas general capabilities only degrade after larger spectral drift.&#x20;
>
> Specifically, the trend in Fig. 3(a) matches the gradual decay of subspace similarity: both change slowly from the early stages because they reflect the model’s ability to answer **specific knowledge**, which i&#x73;**&#x20;sensitive** to even **small parameter shifts**. As a result, accuracy begins to decrease gradually from the start.
>
> By contrast, Fig. 3(b) drops much more sharply. This curve reflects **general abilities**, which are **more robust** and only deteriorate once parameter drift accumulates beyond a certain threshold, producing the sudden drop observed.

---

### Official Review · Reviewer_jfow · 2025-11-01

**Soundness:** 2
**Presentation:** 3
**Contribution:** 2
**Rating:** 6
**Confidence:** 4

**Summary:**

This paper proposes FastEdit, a method designed to address the high computational cost in model editing. By exploiting the low-rank-plus-diagonal (LR+D) structure of hidden representation, FastEdit replaces costly dense inversion with closed-form updates derived via the Sherman–Morrison–Woodbury (SMW) identity. The authors also incorporate a structural prior to improve the robustness of $\mathbf{K}_0$. Experiments show up to $10\times$ end-to-end speedups over prior model editors while maintaining competitive editing efficacy on standard benchmarks.

**Strengths:**

1. The idea of explicitly modeling the editing module with an LR+D structure is intuitive, which unlocks SMW-based closed forms.

2. The mathematical development is rigorous: (i) it motivates LR+D for hidden representations \mathbf{k}, (ii) it derives a closed-form objective and its solution, and (iii) it shows how SMW eliminates cubic-time inversions in practice.

3. The writing is easy to follow, with equations that map directly to the implementation.

4. The approach scales naturally to larger models, addressing a key bottleneck for real-world edits.

**Weaknesses:**

1. While the reported acceleration is impressive, the paper centers its complexity discussion on the inversion term $\left(\mathbf{K}_0\mathbf{K}_0^\top+\mathbf{K}_1\mathbf{K}_1^\top\right)^{-1}$, which may constitute only a fraction of the total runtime in practical pipelines. It is unclear how much of the measured speedup is attributed to this step versus other components (e.g., calculating K_1 and V_1). Releasing code and adding a detailed runtime breakdown would substantiate the claim.

2. The paper does not specify the final choice of the various hyperparameters. Additionally, the joint effects and correlation between them are not yet clear.

**Questions:**

1. Which components dominate the runtime in practice? How large is the measured reduction attributable specifically to replacing inversion with the SMW-based LR+D solver? A runtime analysis would help verify that the claimed acceleration indeed stems from the algorithm.

2. How do the authors tune these hyperparameters (search grid, validation criterion, fixed default)? Is it tuned per dataset/model? The authors are encouraged to provide a more detailed explanation of hyperparameter tuning.

3. In the lifetime model editing, SMW can also be used naturally because $\mathbf{K}_t\mathbf{K}_t^\top$ is a low-rank update. Can the author compare the efficiency of the proposed method?

---

> ### Comment · Area_Chair_zott · 2025-11-12
> **Check and Update Your Comment**
>
> Reviewer jfow,
>
> Please check your comment, which is unrelated to this submission, and correct it asap. Thanks.
>
> Yours,
> AC

---

> > ### Comment · Reviewer_jfow · 2025-11-12
> >
> > Dear AC and authors,
> >
> > I sincerely apologize for mistakenly pasting the wrong review. I will find the right one and fix it within half an hour.

---

> > > ### Comment · Reviewer_jfow · 2025-11-12
> > >
> > > I have fixed the comments. Very sorry again for this issue.

---

> > > > ### Author Response · Authors · 2025-11-26
> > > > **Reply（part3/3）**
> > > >
> > > > **(Q1) While the spectral analysis is novel, the approach of projecting updates is similar to those like AlphaEdit. In addition to empirical results, the authors should provide a more detailed conceptual comparison and discussion.**
> > > >
> > > > We appreciate the reviewer’s comment. While both AlphaEdit and our REVIVE use projection in their update rules, the two approaches differ in \*\*their assumptions about why editing fails, the origin of their protected subspaces, and what the projection is meant to preserve\*\*.
> > > >
> > > > (1) The two methods protect fundamentally different subspaces &#x20;
> > > >
> > > > * AlphaEdit assumes the sequential editing procedure will disrupt the internal knowledge of the pretrained model and builds a knowledge covariance matrix (from 100k factual triples) to extract a null space that avoids knowledge interference. Its protected space therefore comes from \*\*external knowledge statistics\*\*. &#x20;
> > > >
> > > > * Our REVIVE is motivated by a different failure mode—\*erosion of dominant singular directions\*. By examining the spectral structure of the parameters themselves, we find that general abilities reside in \*\*high-energy singular directions\*\*, and we protect this \*\*intrinsic dominant subspace\*\*.
> > > >
> > > > Although both approaches involve projection, the underlying subspaces—hence the motivations—are entirely different.
> > > >
> > > > (2) These differing motivations also lead to **different method designing and thus different performance** .AlphaEdit works well early on by reducing disruption of the preserved knowledge , but its performance still decays in long editing sequences due to noise accumulation.  As we can see in table 1, after 10k edits, the Efficacy Success of **AlphaEdit declines to 62.48**, while **REVIVE equiped version still remains 98.74**, the other editing metrics also reveals the same difference. And for the GLUE performance which represents the model's general abilities, **AlphaEdit drops to zero on all datasets**, while our REVIVE equipped version remains **more than 90% of the unedited model's score**.&#x20;
> > > >
> > > > (3) Our contribution lies in identifying that the sequential editing failure is because of the corruption of the dominant singular directions of weight matrices, introducing a principled way to select a protective subspace, and enabling scalable long-horizon editin\*\*.
> > > >
> > > > In short, although projection appears in both methods, the motivations, protected subspaces, and long-sequence behaviors differ substantially. Our contribution centers on the new failure mechanism and its corresponding spectral perspective, rather than on reusing the idea of projection from AlphaEdit.

---

> ### Author Response · Authors · 2025-11-26
> **Reply（part1/3）**
>
> **(W1) The analysis and method are focused on FFN layers. However, recent work has also explored editing other components, such as attention layers. The authors do not discuss how REVIVE would apply to these new methods.**
>
> We appreciate the reviewer’s thoughtful question.  Although recent methods like PMET\[1] and SADR\[2] discuss how attention behaves during editing, they share a key point: **they interpret attention patterns but do not update attention-layer parameters.&#x20;**&#x4D;ore specifically:
>
> * PMET finds that attention stores little and unstable factual information, and therefore advises against editing attention parameters directly. &#x20;
>
> * SADR regularizes attention responses to avoid over-attending to the edited fact but likewise does not modify attention weights.
>
> Consequently, **current editing methods focus parameter updates almost exclusively on FFN components (e.g., down\_proj), with attention layers serving mainly for behavioral analysis.&#x20;**
>
> We will also explore editing techniques for attention layers in future work, and we have included this point in the *Limitations and Future Work* section.
>
> \[1]  Pmet: Precise model editing in a transformer.AAAI  2024.
>
> \[2]  Revealing and mitigating over-attention in knowledge editing. ICLR 2025.
>
>
>
> **(W2) I have concerns about the effectiveness of the method's heavy reliance on SVD.  On larger models, the dimensionality of the weight matrix will be very high. It is not clear that the SVD can be computed efficiently at that scale.**
>
> We thank the reviewer for the question. Mainstream knowledge-editing methods do not update the full model; they modify only a small set of FFN down\_proj matrices, which prior work shows are the primary loci of editable knowledge. In LLaMA3-8B, for example, only **5 of 32&#x20;**&#x64;own\_proj matrices are updated. REVIVE follows this standard design, so SVD is required only for these few matrices.
>
> In our setup, running `torch.linalg.svd` on a single LLaMA3-8B down\_proj matrix takes about **2.15 seconds**, resulting in roughly **10 seconds** of overhead per REVIVE update. Across a **10,000-edit** , this adds only \~16.6 minutes. For larger models (e.g., LLaMA3-70B), a down\_proj SVD takes about 10.2 seconds, which would translate to \~85 minutes for 10,000 edits—still modest at this scale.
>
> Given that REVIVE improves both editing and GLUE performance by over **50%**, we believe the additional SVD cost is small relative to the performance gains and remains practical even for large LLMs.

---

> ### Author Response · Authors · 2025-11-26
> **Reply（part2/3）**
>
> **(W3):The method employs a hard projection that completely nullifies all update components in the dominant subspace. This might prevent the model from successfully editing certain facts that require a minor change to this core subspace.**
>
> We thank the reviewer for the helpful comments and suggestions. We acknowledge that hard projection can occasionally cause certain edits to fail. However, we believe this impact is limited.
>
> This limited impact is supported by the spectral structure of FFN matrices. Their singular values follow a **long-tailed** distribution, and prior analyses\[1]\[2]demonstrate this by projecting singular vectors directly onto the vocabulary. They find that **tail singular vectors** correspond to **interpretable lexical or factual patterns**, while **top singular vectors** mostly map to **generic, non-semantic tokens**. This aligns with our observation that protecting head singular directions preserves general ability without meaningfully hindering fact-level edits.
>
> Furthermore, Our singular-value–threshold study (Fig. 8) further shows that REVIVE is robust to different threshold choices, reinforcing that these dominant directions contribute little to specific knowledge storage. We appreciate the reviewer’s comment and plan to explore more precise projection strategies to reduce the few early failures.
>
> \[1]Sid Black, *The Singular Value Decompositions of Transformer Weight Matrices Are Highly Interpretable*, 2023 .
>
> \[2]Staats M, Thamm M, Rosenow B. Small Singular Values Matter: A Random Matrix Analysis of Transformer Models\[J]. Neurips, 2025.
>
> **(Q2) The method relies on a static dominant subspace derived from the original model weights. As a significant amount of new knowledge (potentially including foundational facts) is integrated over thousands of edits, is it guaranteed that this initial subspace still accurately represents the model's general abilities?**
>
> We thank the reviewer for the helpful comments, and we apologize if our writing caused confusion — we in fact **recompute the SVD after each batch edit** to adapt to the model’s evolving parameters. And the cost of this extra computation is analysed in the response to W2

---

### Author Response · Authors · 2025-12-03
**Summary for Area Chair(Part1/2)**

During the rebuttal period, we provided detailed responses to all reviewers. Reviewer **qf6z** explicitly acknowledged our clarifications and raised their score, while others did not respond further. To help the Area Chair quickly grasp the core contributions and our responses, we summarize our **contributions ,&#x20;**&#x61;nd the main **strengths** highlighted by reviewers and the key **concerns** raised.

***

## **Summary of Contributions**

Our paper provides a **comprehensive investigation** into why **sequential knowledge editing** leads to **catastrophic model collapse**, and proposes a principled, **plug-and-play** solution that scales sequential editing to an unprecedented **20,000 edits** while maintaining **over 90% editing success** and **minimal general ability loss**.

First, we conduct a detailed **spectral analysis** revealing that a model’s **general abilities** are primarily encoded in a **dominant singular subspace** associated with the largest singular values—directions that are both **essential** and **highly fragile**. Through controlled experiments, we show that perturbations to this subspace cause **immediate degradation** in GLUE performance, and that existing editing methods progressively corrupt these high-energy directions. This allows us to tightly connect **subspace drift** with the observed collapse in both **editing performance** and **general capabilities** during sequential editing.

Second, building on these insights, we introduce **REVIVE**, a simple, modular, and theoretically grounded framework that **explicitly protects** the dominant singular subspace during parameter updates. By decomposing updates in the **SVD-aligned basis** of the weight matrix and filtering out components that interfere with the protected subspace, our method prevents **cumulative degradation** while remaining fully **compatible** with existing editors. Although some reviewers noted similarities to prior projection techniques, our key contribution lies in identifying the **correct subspace to preserve**—one rooted in the model’s intrinsic structure rather than heuristic design. **In short: many can wield the hammer, but the key is finding the right nail.**

Finally, through **extensive experiments** across multiple LLMs, datasets, and editing baselines, we validate the **universality** and **effectiveness** of our method. REVIVE consistently provides stable sequential editing, significantly improves edit success, and preserves **general abilities** far better than state-of-the-art approaches—even under extreme **20k-edit** scenarios.

---

### Author Response · Authors · 2025-12-03
**Summary for Area Chair(Part2/2)**

## **Strengths and Weaknesses from the reviewers**

### **Strengths**

The reviewers(Reviewer x4j5, CBPd, qf6z) agree that the paper demonstrates **strong experimental rigor**. Experiments are **comprehensive**, covering multiple models and tasks—including long-horizon sequential editing—and the **results are consistently stable** and clearly reported. This reflects substantial effort in both engineering and empirical evaluation. The method has clear practical engineering value: REVIVEEDIT is **simple, modular, and plug-and-play**, making it easy to integrate into existing editing frameworks, thus improving robustness in large-scale real-world applications. &#x20;

On the conceptual side, the reviewers(Reviewer jfow, x4j5, CBPd) find the paper’s **spectral explanation insightful**. The work provides a coherent account linking degradation in general abilities to the dominant singular subspace, offering a meaningful theoretical lens for understanding failure modes in sequential editing.&#x20;

Overall, the reviewers acknowledge strong contributions across theoretical explanation, method design, and extensive empirical validation.

### **Weaknesses and Our Responses**

In a nutshell, there are actually two main concerns raised by all the reviewers.

The **first and the main concern** was the perceived similarity between REVIVE and prior projection-based methods such as AlphaEdit, PRUNE, and O-EDIT. In the rebuttal, we clarified that while all methods use projection as a mathematical tool, **the protected subspaces and the underlying motivations are fundamentally different**. To make this explicit, we included the following comparison:

| Method                  | Protected Subspace Source                                 | Target Failure Mode                              | Limitation                                                                     | Difference vs. Our Method                                                             |
| ----------------------- | --------------------------------------------------------- | ------------------------------------------------ | ------------------------------------------------------------------------------ | ------------------------------------------------------------------------------------- |
| **AlphaEdit**           | Feature covariance from **100k external factual triples** | Prevent interference across factual activations  | Subspace is **extrinsic** and tied to sampled data; collapses after \~8k edits | REVIVE protects an **intrinsic** spectral structure derived from model weights        |
| **PRUNE**               | Restrain singular values exceeding original σ\_max        | Avoid overly large updates                       | Magnitude-only filtering ignores singular direction                            | REVIVE selectively filters **harmful directions to the model's general abilities**    |
| **O-EDIT / Delta-Edit** | Past gradient/edit directions                             | Prevent edit–edit interference                   | Only handles conflicts between edits; does not address intrinsic drift         | REVIVE targets **spectral degradation**, a failure mode prior work has not identified |
| **REVIVE (ours)**       | Dominant singular vectors of FFN matrices                 | Collapse due to **dominant subspace corruption** | -                                                                              | Enables **20k+ stable edits**—first method to address this mechanism                  |

This comparison highlights that although projection is a shared primitive, the conceptual foundation and protected subspace in REVIVE differ sharply from prior work. The reviewers accepted that our mechanism-level insight and empirical findings constitute a distinct contribution.

The **second concern** was the computational overhead introduced by repeated SVDs. We clarified that editing methods modify only a small subset of FFN down-projection matrices (e.g., 5 of 32 in LLaMA3-8B). An SVD on such a matrix takes **2.15 seconds** on 8B and **10.2 seconds** on 70B, resulting in roughly **10 seconds overhead per edit**. Even across 10,000 edits, the total additional cost is **\~16.6 minutes** (8B) or **\~85 minutes** (70B). Given that REVIVE yields **over 50% gains in both editing metrics and GLUE performance**.

***

## **Conclusion**

Reviewers acknowledged the strong empirical foundation and the mechanism-driven spectral explanation provided in our work. Concerns about similarity to prior approaches and computational cost were addressed with clear conceptual distinctions and quantitative evidence. Reviewer **qf6z** raised their score after considering the rebuttal. We hope this summary helps the Area Chair gain a concise and accurate overview of our contributions and clarifications.

---

### Note · Authors · 2026-01-05

I have read and agree with the venue's withdrawal policy on behalf of myself and my co-authors.